# Family planning in Pacific Island Countries and Territories (PICTs): A scoping review

**Relmah Baritama Harrington** [1,2]*, **Nichole Harvey** [1], **Sarah Larkins** [1],
**Michelle Redman-MacLaren** [1]

1 College of Medicine and Dentistry, James Cook University, QLD, Australia, 2 Atoifi College of Nursing, Pacific Adventist University, Auki, Solomon Islands

☉ These authors contributed equally to this work.
* relmah.harrington@my.jcu.edu.au

**Data Availability Statement:** All relevant data are within the paper and its supporting information files. Additionally, the data used in this study can be found at the James Cook University repository,

## Abstract

The use of contraceptives for family planning improves women's lives and may prevent maternal deaths. However, many women in low and middle-income countries, including the Pacific region, still die from pregnancy-related complications. While most health centres offer family planning services with some basic contraceptive methods, many people do not access these services. More than 60% of women who would like to avoid or delay their pregnancies are unable to do so. This scoping review identifies and analyses evidence about family planning service provision in Pacific Island Countries and Territories (PICTs), with the aim of better informing family planning services for improved maternal health outcomes in the Pacific. We used Arksey and O'Malley's scoping review guidelines, supported by Levac, Colquhoun and O'Brien to identify gaps in family planning service provision. Selected studies included peer-reviewed publications and grey literature that provided information about family planning services from 1994 to 2019. Publication data was charted in MS Excel. Data were thematically analysed and key issues and themes identified. A total of 45 papers (15 peer-reviewed and 30 grey literature publications) were critically reviewed. Five themes were identified: i) family planning services in the Pacific; ii) education, knowledge and attitudes; iii) geographical isolation and access; iv) socio-cultural beliefs, practices and influences; and v) potential enabling factors for improved family planning, such as appropriate family planning awareness by health care providers and services tailored to meet individual needs. While culture and religion were considered as the main barriers to accessing family planning services, evidence showed health services were also responsible for limiting access. Family planning services do not reach everyone. Making relevant and sustainable improvements in service delivery requires generation of local evidence. Further research is needed to understand availability, accessibility and acceptability of current family planning services for different age groups, genders, social and marital status to better inform family planning services in the Pacific.

at: https://research.jcu.edu.au/data/published/
07d0933d17f3070f5e5e38da36bfdfff/.

**Funding:** The authors received no specific funding
for this work.

**Competing interests:** The authors have declared
that no competing interests exist.

## Introduction

In 2017, more than 800 women around the world died every day from preventable causes
related to pregnancy and childbirth complications [1–3]. Many of these deaths occurred in
low and middle income countries (LMICs) including those in the Pacific region [4]. Adoles-
cent females and women living in rural areas face higher risks of unintended pregnancies,
complications and death compared to other women [5, 6]. Health education and contracep-
tion knowledge, with access to appropriate health services can empower women and men to
make informed decisions about their reproductive choices [7]. Increasing evidence also
showed that empowering women empowers humanity: families are healthier, and better edu-
cated, and economies also grow faster [8]. Ensuring access to sexual and reproductive health
services (SRH), including family planning (FP), is a fundamental human right and can be a
cost-effective approach to prevent pregnancy complications that lead to maternal deaths [9,
10].

The United Nations (UN) population conferences in Rome (1954) and Belgrade (1965)
highlighted the issue of FP in light of rising populations and the threat of mass starvation [11,
12]. Population control policies were created, but, these policies failed to address the dimen-
sions of social inequality in terms of human rights [11]. Subsequent UN population confer-
ences, including the Committee on the Elimination of Discrimination against Woman, 1979;
Bucharest 1974; the International Conference on Population, Mexico, 1984; and World
Human Rights, Vienna 1993 continued the discussion with greater contributions from
women, religious groups and less developed countries [13]. The need to change approaches to
population control and its relationship to development became evident. The status of men and
women in the family and society were then fully realised in FP discourse. This recognition led
to the initiation of the reproductive and sexual health rights concept as an alternative to the
former narrowly FP program approach [9, 14].

At the International Conference on Population and Development (ICPD), Cairo 1994, the
international community reached an unprecedented global consensus on population issues
and the concept was endorsed by 179 countries. The ICPD Program of Action (ICPD PoA) set
out a series of priority issues including among others, population and development, gender
equality and equity, reproductive health and rights and adolescents and youth [15]. The Beijing
Declaration (1995) further supported the notion of gender equality and the empowerment of
women everywhere [8]. Central to the ICPD PoA [16] is the attainment of reproductive rights
and reproductive health. All countries are expected to ensure that comprehensive reproductive
health services including FP are accessible, affordable and acceptable to all individuals through
the Primary Health Care (PHC) system. This comprehensive package includes: (i) FP counsel-
ling, information, education, communications and services; (ii) education and services for safe
pregnancy, childbirth and postnatal care; (iii) prevention and appropriate treatment of unsafe
abortion; (iv) treatment of reproductive tract infections and appropriate information educa-
tion and counselling for sexually transmitted infections (STIs), including human immunodefi-
ciency virus (HIV); and (v) promoting sexual health [17, 18]. Family planning is a component
of reproductive health that has a strong natural link with the other four program components
and is a pre-requisite for achieving all other sustainable development goals [19]. Family plan-
ning, according to the World Health Organization (WHO), allows individuals and couples to
anticipate and attain their desired number of children, and the spacing and timing of their
births, through the use of modern contraceptive methods [20].

The review of the ICPD PoA goals in the United Nations Funds for Population Activities
(UNFPA) Pacific progress reports [3, 18, 21] identified that these goals remain relevant.
Although significant progress was made, greater action is needed. The Millennium

Development Goals (MDGs: 2000–2015), provided the global framework following the ICPD, with Targets 5a "to improve maternal health" and 5b "universal access to reproductive health" [22]. The Sustainable Development Goals (SDGs: 2015–2030) continue to provide the platform for this global agenda [11, 23]. The third SDG, good health and wellbeing, espouses inclusivity and not leaving anyone behind, regardless of their age. Targets 3.1 and 3.7 respectively supported reproductive health and FP, aiming to reduce the global maternal mortality ratio to below 70:100,000 live births and provide universal access to SRH services. Underpinning the provision of services in the SDG era is the notion that services focus on the marginalised and most at risk groups such as adolescents and the principle of "informed free choice" governs FP programs [18, 24]. This shift in global thinking has had major policy and programing implications for reproductive health in the Pacific region. The diversity of the Pacific region reflects the complexities in establishing a one-size-fits-all FP program. Contraceptive needs may not be met due to limited, inconvenient or inappropriate services, cultural factors or religious beliefs [25]. Reaching these SDG targets by 2030 will require context relevant programs and policies to be incorporated into each country's national strategic plan [22, 26].

For the purpose of this review, the Pacific region comprises of 21 Pacific Island Countries and Territories (PICTs) dispersed throughout the Pacific, often referred to as 'large ocean states' [27]. These countries represent an enormous diversity in physical geography and culture, languages and social-political organisations, population size and development, and are classed in three main ethnic sub-groups. Melanesia includes Fiji, New Caledonia, Papua New Guinea (PNG), Solomon Islands, and Vanuatu. Polynesia includes American Samoa, Cook Islands, French Polynesia, Guam, Niue, Samoa, Tokelau, Tonga, Tuvalu, Wallis and Futuna, while Micronesia includes The Federated States of Micronesia, Kiribati, Marshall Islands, Nauru, and Northern Mariana Islands [14]. With small populations and land areas amongst vast ocean spaces, limited resources and a narrow economic base affect these countries—most rely heavily on official development assistance from higher income countries and international partners [27]. Modern FP programs were introduced in PICTs in the 1960s to promote population reduction and socio-economic development as well as to improve women's and children's health [28]. At the ICPD, the Pacific community accepted the PoA and recognised sexual and reproductive health rights (SRHR) as fundamental to human rights [29], and have since committed to improving the reproductive health of their people [30]. While significant global achievements have been made, such as decreased maternal and infant mortality ratios, improved access to contraception, falling fertility rates, and increased life expectancy, progress for improved SRH in the Pacific has been slow and inconsistent [8]. With low Gross National Income per capita (<$3,000) and high population growth rates (>2.0%), Pacific countries such as Solomon Islands, Vanuatu and PNG have inadequate resources to support current population growth [10, 14]. The realisation of rights and social protection for vulnerable groups such as women and children, the elderly, youth and people with disabilities is inadequate. Integrated and comprehensive approaches to achieving SRHR across the region are yet to be fully established. Pacific countries are in various stages of implementing the "Family Life Curriculum" [3, 31] in schools and establishing youth friendly services. However, the integration of population issues into education systems is still under development. Enabling women to enjoy full participation in political and economic life remains a challenge and gender-based violence is prevalent in many PICTs [21].

Contraceptive Prevalence Rates (CPR) and 'unmet need for FP' have been used as indicators to measure the uptake of contraceptives in FP programs. The CPR in the Demographic and Health Surveys (DHS) and Multiple Indicator Cluster Surveys (MICS) is defined as the percentage of women age 15–49 years currently married or in union, who are using or whose partner is using any contraceptive method at the time of the survey. 'Unmet need for FP' is the

percentage of married women or women in union who want to stop or delay childbearing but are not using any contraception [32]. Both indicators were used to determine if women or couples were taking any action or using any method to delay or avoid getting pregnant [23]. In the Pacific, the CPR is estimated to be 18–48%, well below the 62%, average for Low Income Countries [14]. There is also a relatively high level of unmet need for FP (8–46%) when compared to global estimates of less than 10% unmet need for FP [23]. The persistently high total fertility rate of 3–4% compared to 1% globally reflects the low CPR and high unmet need for FP in PICTs [10, 24, 33].

Before the 1994 Cairo conference, FP was delivered within the Maternal Child Health and Family Planning (MCH/FP) context, and primarily targeted married women. After 1994, the integration of these services into more holistic and comprehensive approaches including SRH was considered [28, 34]. More recently, this MCH/FP approach included the broader context of Reproductive, Maternal, Neonatal, Child and Adolescent Health (RMNCAH) to reach the unmarried and adolescents [35, 36].

However, many adolescents, women and men do not access these services [36]. Little is known about how FP services are accessed and provided at health facilities throughout the Pacific [37, 38]. For this reason, a scoping review was conducted to map key concepts from a wide range of literature to identify gaps to inform further research and for improved FP services in the Pacific region [39–41]. To understand how FP services have been implemented in PICTs, we reviewed and synthesised the literature on provision of FP services in PICTs and the successes and challenges of service implementation. This review focused on two research questions:

1. How have FP services been implemented in PICTs between 1994 and 2019?

2. What are the successes and challenges in providing FP services in PICTs?

For the purpose of this review, 'FP service' refers to any service within SRH care that provides contraception and counselling services purposely to prevent or delay pregnancy.

## Method

Scoping reviews are useful in health research, to map key concepts and identify literature gaps. They are particularly useful when little is known about a topic. We followed the guidelines for conducting scoping reviews established by Arksey and O'Malley [39], Levac, Colquhoun and O'Brien [40] and the Joanna Briggs Institute [42], to summarise peer reviewed journal papers and relevant grey literature including government and organisational reports. This review follows an unregistered protocol (S1 Appendix) developed prior to conducting the study and structured consistent with the Preferred Reporting Items for Systematic Reviews and Meta-analysis Protocols extension for Scoping Reviews (PRISMA-ScR) checklist [41].

### Eligibility criteria

Original research studies of all designs including grey literature conducted in PICTs were considered.

To be included in the review, papers had to meet the following inclusion criteria (S2 Appendix):

1. Report on the FP service component of the SRH Care Services in PICTs or world regions that include countries in the Pacific;

2. Report on the successes/enablers and challenges/barriers to FP service provision;

3. Published in the English language; and

4. Published between 1994 and 2019 to capture the 1994 ICPD focus on the global commitment strategy for universal access to SRH including FP, and also encompass the period of the MDG and commencement of the SDGs. The PICTs also signed an agreement to ICPD in 1994 and this guides their progress towards achieving targets for the SDGs.

Papers discussing SRH and STIs that did not include aspects of FP were excluded, as were papers focused on surgical termination of pregnancy as a form of FP, antenatal care and pregnancy services.

## Information sources and search

Informed by the research questions, we determined keywords (S3 Appendix), before constructing search strategies. University librarians assisted to review and confirm search strategy drafts for electronic databases and grey literature searches, which were then refined through author group discussion. Scopus, MEDLINE (Ovid), CINAHL and PsycINFO databases were searched using key words and database-specific subject headings to identify relevant studies. These databases were chosen as they provide most relevant peer reviewed articles about family planning in the Pacific. The search strategy was adapted for each database. The search was conducted between 2018 and 2019. The final search strategy for MEDLINE (Ovid) can be found in S4 Appendix. Searches were performed for published and unpublished work on Google Scholar, organisational websites (WHO, UNFPA, United Nations International Children's Emergency Fund, UN), Pacific-based journals and reports (i.e. Pacific Journal for Reproductive Health, South Pacific Commission) including government reports such as Demographic and Health Surveys. Organisational websites and Pacific-based journals and reports were selected to augment papers identified by electronic sources, as these were known to report on reproductive health services including FP in PICTs.

## Selection of sources of evidence

All papers were imported into Endnote bibliographic software and duplicates removed. Two authors (RH and MRM) performed the initial screening. Papers that did not clearly meet the inclusion criteria were reviewed by the other two authors (NH and SL) before a decision was made to either include or exclude the papers from the review. We extracted data using a MS Excel spreadsheet designed for this review to capture information about the study characteristics (publication year, country of study, study focus, design), and a data extraction sheet (Table 1) where key findings of FP services were recorded (S5 Appendix). The PRISMA flow chart in Fig 1 shows the procedure for selecting papers for inclusion.

## Data charting process

Full-text papers that met the inclusion criteria were thoroughly read to capture relevant information required in the review [39, 40]. Findings were analysed using content analysis and synthesised using a thematic, narrative approach [42, 44]. During this stage, decisions about what information should be recorded from the primary studies were made using an iterative process [40]. Given that limited peer reviewed research was conducted on FP service provision in the Pacific region, studies were not excluded on quality grounds but included purposely to map available evidence as consistent with the scoping review methodology. However, the overall quality of the included studies was limited.

**Table 1. Data extraction sheet.**

| Authors/Date/Location | Focus | Publication type | Methods/Sample | Key Findings |
|---|---|---|---|---|
| **Brewis et al., 1998** [45] Samoa | Assess family planning (FP) acceptance | Original research | Qualitative (n = 155) women 15–49 years | • Awareness and use of contraception have markedly increase in both rural and urban areas |
| | | | | • Availability and accessibility to contraceptives reportedly high |
| | | | | • Contraceptives made accessible and affordable for rural and urban woman by government |
| | | | | • Younger women desired larger families |
| | | | | • FP needs further investigation to be clearly understood |
| **Burslem et al., 1998** [46] **Solomon Islands** | Teenage pregnancies and sexually transmitted infections | Descriptive research | Questionnaire (n = 266) high school students. Focus group (n = 12) women and girls. Interview (n = 24) college students, pregnant single mothers | • FP services unavailable to unmarried people regardless of age |
| | | | | • Poor knowledge about FP services |
| | | | | • Poor access to condoms |
| | | | | • A sympathetic health worker is needed |
| **Cammock et al., 2017** [47] **Fiji** | Socioeconomic and cultural contexts | Original research | Cross sectional study (n = 212, women of childbearing age) | • FP service not culturally-sensitive |
| | | | | • Cost of service and language are main barriers |
| | | | | • Need culturally relevant services |
| **Daube et al., 2016** [48] **Kiribati** | Knowledge, use and barriers to contraceptive uptake for women and men | Descriptive research | Mixed method (n = 500) women (15–49 years) and men (15–54 years) | • Unsuitable service delivery |
| | | | | • Barriers include, not interested in FP, knowledge gaps, personal reasons, family & social obligation |
| **Davis et al., 2016** [49] **Cook Islands, Solomon Islands, Fiji, Vanuatu, Papua New Guinea** | Attitudes and belief regarding benefits, challenges, risks and approaches to male involvement in reproductive health | Descriptive research | Qualitative study (n = 17) senior Maternal Child Health policy makers and practitioners | • FP services not focused on men/men are not involved |
| | | | | • Perceived challenges–socio-cultural norms, physical layout of clinic, health workers attitudes and work loads |
| | | | | • To engage boys and men early in the life cycle |
| **Hayes and Robertson, 2012** [50] Pacific Island Countries | Current status and prospects for repositioning FP on the development agenda | Report | Not provided | • Distribution and dispensing of contraceptive in the Pacific mainly through Government-operated health facilities, Family Health Associations and Private pharmacies or doctors in private practice Generally free services |
| | | | | • Most Pacific countries incorporate reproductive health including FP into national and subnational development plans |
| | | | | • CPR ranged from 17–49% in PICTs. Method of measurement may not be comparable and accurate |
| | | | | • Recently introduced DHS in PICTs |
| **House and Ibrahim, 1999** [51] Pacific Island Countries | Adolescent birth rates | Discussion paper | Not provided | • Focused on adolescent services and no special attention to older women's reproductive health needs |
| | | | | • Inconvenient and unsatisfactory services |
| | | | | • Higher fertility and unmet needs among women aged over 35 |
| | | | | • Rising reproductive health status of adolescents, resulted in declining fertility rates over three decades |

(*Continued*)

**Table 1.** (Continued)

| Authors/Date/Location | Focus | Publication type | Methods/Sample | Key Findings |
|---|---|---|---|---|
| **House and Katoanga, 1999 [28] Pacific Island Countries** | Reproductive health and FP in Pacific island countries | Discussion paper | Not provided | • Before Cairo conference MCH/FP centred on the pregnant mother and her child |
| | | | | • FP targets married women. In practice MCH/FP implemented separately from other SRH components (STI/HIV) |
| | | | | • Challenges: raising awareness, identifying |
| | | | | • priorities for adolescents SRH needs and integrating services into more holistic and comprehensive approach |
| | | | | • Success: reproductive health training program was established in Fiji |
| **Kennedy et al., 2011 [2] East Asia and Pacific Island Countries and Territories** | Adolescent fertility- current use, knowledge and access to FP information and service | Review | Not provided | • Married and unmarried adolescents have less access, low use and high-unmet need for contraceptives. |
| | | | | • Adolescents lack knowledge about services compared older women |
| | | | | • Concerns about gender of health providers, poor geographical access and financial barriers |
| **Kennedy et al., 2013a [10] Vanuatu and Solomon Islands** | Health, demographic and economic consequences of reducing unmet need for FP | Intervention research | Using demographic modelling | • Increasing investment in FP could contribute to improved maternal and infant outcomes and substantial public savings and lower dependency ratio |
| **Kennedy et al., 2013b [5] Vanuatu** | Service providers' perceptions of youth-friendly SRH services in Vanuatu | Original research | Qualitative study (n = 66 Focus group) with 341 male and female adolescents. (n = 12 interviews) with policy makers and service providers | • Government provides most SRH service. Small number of youth facilities provided by non-government organisations |
| | | | | • Service focused mainly on STIs and HIV |
| | | | | • Adolescents lack knowledge about prevention of pregnancy, condom use, puberty and sexual relations; early sexual debut |
| | | | | • Need friendly service providers and context-specific strategies |
| **Kennedy et al., 2014 [52] Vanuatu** | SRH information preferences of adolescents in Vanuatu | Original research | Qualitative study (n = 66 Focus group) with 341 male and female adolescents. (n = 12 interviews) with policy makers and service providers | • Adolescents mostly access the service to seek information or advice |
| | | | | • Non-government services more accessible than government facilities |
| | | | | • Barriers include socio-economic norms and taboos |
| | | | | • Lack of confidentiality and privacy |
| | | | | • Schools an underutilised source of information. |
| | | | | • Need a wide range of media sources of SRH information |
| **Kenyon and Power 2003 [34] Pacific Island Countries** | Getting the basics of FP in the Pacific region | Discussion paper | Not provided | • Pacific health centres traditionally operate a once a week session for FP. This is likely to be inconvenient for many clients |
| | | | | • No privacy in clinics, confidentiality easily breached in small village clinics, |
| | | | | • Health worker attitudes (negative) |
| | | | | • Outdated population policies/no policies/ policies lack details and coordinating structure/policies that emphasise approaches to FP not shown to be effective |
| | | | | • Socio-cultural values & beliefs |

*(Continued)*

**Table 1.** (Continued)

| Authors/Date/Location | Focus | Publication type | Methods/Sample | Key Findings |
|---|---|---|---|---|
| **Kiribati Demographic health survey, 2009** [53] Kiribati | Contraceptive knowledge, use, attitudes and sources | Report | Mixed method: 1,978 women aged 15–49, 1,135 men aged 15–54 | • Contraceptive prevalence rate– 22% (married women), 16.5% (all women) |
| | | | | • Government/public sector strategically important in providing service through health facilities. Few use private sectors. Others source contraceptives from relatives overseas. Service offered for free |
| | | | | • Challenges in contraceptive use include the desire for many children and religious prohibition |
| **Kura et al., 2013** [54] **Papua New Guinea** | Male involvement in sexual reproductive health (including FP) | Original research | Mixed method, 122 married men aged 21–44 years | • FP clinic services are usually female oriented; men are never targeted on awareness/education on safe motherhood initiatives |
| | | | | • Inadequate services for men, male literacy also contributed to men's participation |
| | | | | • Challenges: illiteracy, inadequate knowledge (importance and benefits of FP), cultural factors, lack of appropriate services. Other factors include wanting more children and fear of religious condemnation |
| **Lee, 1995** [55] Pacific Island Countries | Assess current situation in reproductive health and FP and the way forward | Discussion Paper | Not provided | • Reproductive health and FP are an integral part of MCH/FP framework and focuses on pregnancy and contraception |
| | | | | • Services confined to married women and narrowly focused. Do not address needs of special groups like teenagers and women over 40 years |
| | | | | • Sexually transmitted infections are separate programs from MCH |
| | | | | • Low male and adolescent participation |
| **Lincoln et al., 2018** [56] Fiji | Identify the level of knowledge, attitudes and practices of FP among women of reproductive age | Descriptive research | Qualitative cross-sectional study, 325 women (15–49 years) | • Health centres were the primary sources of in-depth knowledge and awareness regarding contraceptive use compared to other highly influential initiatives |
| | | | | • Barriers to contraceptive use include religious beliefs, cultural beliefs, gender disparities, the need for regular visits to health centres |
| | | | | • Ideal number of desired children in families is between 3 and 5 |
| | | | | • Way forward–greater gender equality, programs to address issue by describing the number of children in an ideal family unit |
| **Marshall, 2017** [57] Kiribati | Strengths and gaps of SRH services | Report | Mixed method (n = 14) community clinics and staff | • Basic FP service provided at most community clinics |
| | | | | • No FP guidelines, lack of standardisation of care across all clinics |
| | | | | • Staff need further education to increase knowledge, confidence and skills to enable contraceptive choices |

(*Continued*)

**Table 1.** (Continued)

| Authors/Date/Location | Focus | Publication type | Methods/Sample | Key Findings |
|---|---|---|---|---|
| **Marshall Islands Demographic health survey, 2007** [58] | Contraceptive use, knowledge, attitudes and behaviour | Report | Mixed method: 1,626 women aged 15–49 1,055 men aged 15–54 | • Contraceptive prevalence rate– 45% (married women), 37% (all women) |
| | | | | • Government is the main source of modern contraception. Services provided for free |
| | | | | • Almost universal contraceptive knowledge for men and women |
| | | | | • Common reasons for non-use are fear of side-effects, loss of fertility and desire for more children |
| **Mody et al., 2013** [59] Asia-Pacific **Countries** | Impact of strategic partnership programs to improve evidence-based guidance | Program description | Multiple methods: Sample not provided | • Key informants who provide information are often program administrators who may not be aware of the actual use of FP materials in the clinics |
| | | | | • Evidence based tools were used to improve training curriculum and materials in Pacific Islands and Territories |
| | | | | • FP guidelines and tools only effective if supplies to meet the increased demand are available |
| **Morisause et al., 2017** [60] Papua New Guinea | *Contraceptive prevalence and barriers to using modern contraception* | Descriptive research | Mixed method (n = 193) women of childbearing age 15–49 years | • Service not culturally accessible |
| | | | | • Village health workers discourage use of contraception |
| | | | | • Low contraceptive prevalence, high unintended pregnancies and unmet need |
| | | | | • Lack of knowledge, staff attitudes, costs, stock availability |
| | | | | • Worried about side-effects, use traditional methods |
| | | | | • Husband/partner opposition, clinic too far |
| **Naidu et al., 2017** [61] **Fiji** | Knowledge, attitudes, practices and barriers to safe sex and contraceptive use | Descriptive research | Cross-sectional study (n = 1490) of rural women aged 18–75 years old who present to sexual reproductive health outreach sessions | • Unmarried people had difficulties accessing service |
| | | | | • High knowledge about pregnancy and how to avoid it (>80%, but low knowledge about the practicalities of contraception (43%) |
| | | | | • Higher education level of women does not correlate with knowledge about emergency contraception and condom use and pregnancy prevention |
| | | | | • Barriers: partner disagreement, lack of contraceptive knowledge |
| **Nauru Demographic health survey, 2007** [62] | Information on contraceptive use, knowledge and attitudes pertaining to contraception | Report | Mixed method: 667 women and 653 men aged 15–49 | • Contraceptive prevalence rate– 36% (married women), 27% (all women) |
| | | | | • FP service not integrated with other reproductive health services |
| | | | | • Wide knowledge of condom use |
| | | | | • High use among younger women and currently married men and lower use in women 35 years and older. Men are reported to use 4 male methods |
| | | | | • Lack emphasis on discussing FP issues due to lack of home visits |
| | | | | • Desire for more children is the common reason for non-use |

*(Continued)*

**Table 1.** (*Continued*)

| Authors/Date/Location | Focus | Publication type | Methods/Sample | Key Findings |
|---|---|---|---|---|
| **Papua New Guinea Demographic health survey, 2016–18** [63] | Awareness and use of FP methods | Report | 19,200 households selected from 800 census units. Women and men aged 15–49 selected for individual interviews | • Contraceptive prevalence rate– 37% (married women), 33% (all women) |
| | | | | • Health facilities common places to source services and contraception. Free services from public sector |
| | | | | • Married educated women most users of service Men and adolescents have less access |
| | | | | • Common challenges: lack of knowledge, want more children, side-effects, hard to get methods and religion |
| **Raman et al., 2015** [6] **Solomon Islands** | Barriers to adolescent SRH service provision | Descriptive research | Mixed method (n = 147) teachers, school principals, youths & health workers | • Services are theoretically available, but some services may be inaccessible due to cultural beliefs |
| | | | | • Unmarried people may not be offered contraception |
| | | | | • Lack of clarity in health workers role for adolescent reproductive health programs, social norms, shortage of resources (understaffing), lack of incentives, ambivalent attitudes, knowledge gaps |
| | | | | • Inadequate training for adolescent sexual reproductive health services |
| **Robertson, 2007** [36] **Pacific Island Countries** | Repositioning FP as an integral development strategy | Discussion paper | Not provided | • Global waning of FP services and emerging threat from HIV/AIDS |
| | | | | • Emphasis on FP diminishes |
| | | | | • High total fertility rate |
| | | | | • Under-reporting of contraceptive prevalence rates, no information on unmet need is available |
| | | | | • DHS data not available in most Pacific countries, prior to 2016 |
| **Roberts, 2007** [64] | Evaluation—design, efficacy and effectiveness of MIRH | Program description | Male workers: Solomon Islands (n = 16), Fiji (n = 21) | • Concept of male involvement in reproductive health well received in Pacific countries but services lack strategies to deal with sensitivities in sexual health issues |
| **Fiji and Solomon** | | | | • Need to measure unmet need for contraceptives |
| **Islands** | | | | • Contraceptive prevalence rates need to be validated through demographic health surveys or related surveys in order to monitor progress |
| **Rowling et al., 1994** [65] | Family planning knowledge, attitudes and practices among married men and women of reproductive age | Descriptive research | Mixed method: (n = 150) women 15–49 years, (n = 90) male 15–54 years | • FP service focuses on married couples, not available to unmarried couples regardless of age |
| | | | | • Women access service more than men |
| **Solomon Islands** | | | | • Poor knowledge about reproduction |
| | | | | • Beliefs, cultural norms and distance influenced use of service and contraception |

(*Continued*)

**Table 1.** (Continued)

| Authors/Date/Location | Focus | Publication type | Methods/Sample | Key Findings |
|---|---|---|---|---|
| **Samoa Demographic health survey, 2014** [66] | Contraceptive use, knowledge, attitudes, sources and attitudes | Report | Mixed method: 4,805 women aged 15–49 and 1,669 men aged 15–54 | • Contraceptive prevalence rate– 27% (married women), 17% (all women) |
| | | | | • Government sector is the main source of provider |
| | | | | • Contraceptives are free |
| | | | | • Women had more access than men |
| | | | | • Knowledge increase over the last 5 years, almost the same in both men and women |
| | | | | • Challenges: respondents and husband/ partner opposition, religious beliefs, method related, health concerns and wanting many children |
| **Solomon Islands National Statistics Office, 2015** [67] **Solomon Islands** | Information on fertility, FP, infant and maternal mortality | Report | Mixed method: women 15-49years (n = 6226), men 15 years and above (n = 3591) | • Contraceptive prevalence rate– 29% (married women), 21% (all women) |
| | | | | • Contraception mostly provided in government/public sectors, few by private, faith based and non-government organisations. Service is mostly free |
| | | | | • High unmet need in rural than urban areas, high fertility rate and mortality rates |
| | | | | • Unmarried and young women and men have less access |
| **Tonga Demographic health survey, 2012** [68] | Contraceptive knowledge, use, attitudes and sources | Report | Mixed method: 3,068 women and 1,336 men 15–49 years | • Contraceptive prevalence rate– 34% (married women), 20% (all women) |
| | | | | • Women in rural areas more likely to use a method than urban women. Knowledge high among currently married women and men |
| | | | | • The government provided most services and contraception. Condoms distributed in clinic through peer educators. Services are provided free |
| | | | | • Reasons for non-use include: fear of side-effects, desire for many children, health concerns, husband/partner opposition and religious prohibition |
| **Tuvalu Demographic health survey, 2007** [69] | Contraceptive use, knowledge, attitudes and behavior | Report | Mixed method: 850 women and 428 men aged 15–49 | • Contraceptive prevalence rate– 30.5% (married women), 23.1% (all women) |
| | | | | • Source mainly from public/government sector. |
| | | | | • High knowledge in all women and men including unmarried sexually active men |
| | | | | • Reasons for intending to use contraceptives include fear of side-effects, desire for more children, health concerns, opposition by respondent and religious beliefs (fear of side-effects and desire for more children are common reasons) |

(*Continued*)

**Table 1.** (Continued)

| Authors/Date/Location | Focus | Publication type | Methods/Sample | Key Findings |
|---|---|---|---|---|
| **UNFPA, 2004 [18]** Pacific Island Countries | Progress at 10 years after ICPD | Report | Not provided | • Pacific countries remain highly supportive of ICPD but progress in implementing its recommendations varied across the region |
| | | | | • Some countries developed population policies others have taken steps to prepare but not reaching implementation stage |
| | | | | • Less progress in integration of population into sector plans and strategies |
| | | | | • Most countries now capable of conducting a population census but the capacity to process, analyse, and interpret census survey results from policy perspective remains limited |
| **UNFPA, 2009 [3]** Pacific Island Countries | Progress at 15 years after ICPD | Report | Not provided | • SRH not well coordinated and holistic due to vertical, fragmented and under resourced nature of programs. |
| | | | | • FP not reaching groups who need it |
| | | | | • High unmet need in older women, lifetime fertility remains above 4 children per woman in several countries |
| | | | | • Programs require renewed political support and innovative strategies to meet needs of disadvantaged groups |
| | | | | • Conduct more socio-cultural research on factors inhibiting use of FP |
| **UNFPA, 2014a [21]** Pacific Island Countries | Progress at 20 years after ICPD | Report | Not provided | • Progress made but pace and extent varied greatly between countries |
| | | | | • Integrated and comprehensive approach to achieving SRH rights yet to be fully established |
| | | | | • Integration of population issues into education systems still under development |
| | | | | • High costs of transport because of remoteness of many communities, a significant barrier |
| | | | | • Effective stakeholder engagement and partnerships reported as common facilitators by governments |
| | | | | • To devote resources to research and understand behaviours of Pacific peoples so that programs on STIs, contraception, and FP are based on best evidence |
| **UNFPA, 2014b [14] 15 Pacific Island countries** | Report—Summary of updated population and development profiles | Report | (n = 6) reproductive health program officers | • Social and heterogeneous culture in the Pacific |
| | | | | • Challenges differ among countries |
| | | | | • Very religious, sensitive issues challenging to discuss |
| | | | | • Weak statistics, high unmet need, high total fertility rate, low contraceptive prevalence rate below 62% average for developing countries |

*(Continued)*

**Table 1.** (Continued)

| Authors/Date/Location | Focus | Publication type | Methods/Sample | Key Findings |
|---|---|---|---|---|
| **UNFPA, 2015a** [70] Kiribati | Existence and use of relevant SRH services, policies and laws (rights) | Report | Mixed method: health facilities visited (n = 16). Interviews and focus group discussions with (n = 8) senior Ministry of Health officers, medical assistants and relevant non-government officers | • Service relies heavily on development partners funding, sustaining progress is a challenge. SRH policy still in draft<br><br>• Need to improve integration of service for both men and women. Poor service to outer islands<br><br>• A signatory to the international health regulations<br><br>• FP services observed to be available and accessible<br><br>• Issues include: understaffing, outdated policies/guidelines, inadequate reporting systems, fiscal and geographical challenges in outer islands |
| **UNFPA, 2015b** [71] Samoa | Existence and use of relevant SRH services, policies and laws (rights) | Report | Mixed method: health facilities visited (n = 11). Consultation/interview with government and non-government health service providers, managers and technical advisors (n = 33) | • Has policy and remains committed to upholding sexual reproductive health rights. All health facilities provide FP service<br><br>• Samoa invests in youth focused programs/ infrastructure<br><br>• Challenges include: cultural and attitudinal barriers at all levels (individual/ communities; village/church leaders; school management committees; government ministries and service providers) and young people limited access to contraception |
| **UNFPA, 2015c** [72] **Solomon Islands** | Existence and use of relevant SRH services, policies and laws (rights) | Report | Mixed method: interview with the health sector officer (n = 1), and non-government officers (n = 6) | • Contraceptives provided in most public sectors, few in private/non-government and faith-based organisations. Lack of integration in all SRH services<br><br>• No SRH policy available but uses the country's Reproductive Health Strategy Implementation Plan 2014–2016, HIV policy, and multi-sectoral strategic plan 2005–2010.<br><br>• Service less accessed by younger women and adolescents. Poor service delivery to outer islands<br><br>• Mixed progress in incorporating gender and rights into SRH agenda<br><br>• Economic issues, cultural and fiscal constraints, understaffing. Outdated policies and guidelines, inadequate health infrastructures, and poor reporting system challenges progress |

(*Continued*)

**Table 1.** (Continued)

| Authors/Date/Location | Focus | Publication type | Methods/Sample | Key Findings |
|---|---|---|---|---|
| **UNFPA, 2015d** [73] Tonga | Existence and use of relevant SRH services, policies and laws (rights) | Report | Mixed method: health facilities visited (n = 14). Key informant interviews Ministry of Health (n = 11), non-government organisations (n = 4). Focused group discussions (n = 4) | • All facilities assessed provide range of SRH services (in clinic or outreach) including FP. Services are free |
| | | | | • No current SRH policy but National Integrated SRH Strategic Plan 2014–2018 guides SRH program |
| | | | | • current SRH policy but National Integrated SRH Strategic Plan 2014–2018 guides SRH program |
| | | | | • Achieved mixed progress to incorporating gender and rights |
| | | | | • Challenges to improved access: understaffing, outdated policies; preventing stock outs; no mentoring programs to monitor skills retention. Others include: geographical isolation, economic, cultural and fiscal constraints |
| | | | | • Actively conducted outreach programs through "settings approach" (schools, villages, workplaces, churches, daily talk back shows) |
| **UNFPA, 2015e** [74] Vanuatu | Existence and use of relevant SRH services, policies and laws (rights) | Report | Mixed method: key informant interviews: Ministry of Health service managers/providers (n = 25); non-government organisations (n = 5); partners (n = 4) | • Committed to upholding human rights of its citizens, evidence through national constitution and signing international conventions and treaties |
| | | | | • Reproductive health policy (2008) and strategy (2008–2010) is the guiding document for delivery of STI/HIV and FP |
| | | | | • health policy (2008) and strategy (2008–2010) is the guiding document for delivery of STI/HIV and FP |
| | | | | • Higher-level health facilities provide comprehensive range of SRH. Aid posts reported not meeting FP promotion |
| | | | | • Challenges: young and growing youth population; understaffing, outdated policies, inadequate reporting systems/processes |
| **UNFPA, 2019** [75] Pacific Island Countries | The State of Pacific's RMNCAH workforce | Report | Not provided | • Most countries have sufficient nurses to meet need for RMNCAH care but shortage of nurse-midwives |
| | | | | • Most have official policy to access RMNCAH care but out of date and not fully costed |
| | | | | • Barriers to service integration–staff shortage and need for further training |
| | | | | • Gender barriers significant, concerns about confidentiality in small settings |
| | | | | • Integration of youth-friendly RMNCAH services rare in the region |

(*Continued*)

**Table 1.** (Continued)

| Authors/Date/Location | Focus | Publication type | Methods/Sample | Key Findings |
|---|---|---|---|---|
| **Vanuatu Demographic health survey, 2013** [76] | Contraceptive use, knowledge, attitudes and sources | Report | Mixed method: 2,508 women and 1,333 men aged 15–49 | • Contraceptive prevalence rate—49% (married women), 38% (all women) |
| | | | | • The government sector is the primary source of service and contraception, service is generally free |
| | | | | • Use is high among currently married women and high wealth quintile |
| | | | | • Knowledge is high among currently married and those with 3 to 4 children |
| | | | | • Reasons for not intending to use: fear of side effects, opposition from respondent/ husband/partner, fertility reasons and lack of knowledge |
| **White et al., 2018** [77] **Cook Islands** | Social and contextual factors that inform contraceptive knowledge, attitudes | Original research | Qualitative study (n = 10) women who were mothers before aged 20 years old | • Access to contraception is not sufficient, rates of adolescent pregnancy remains the same despite availability of services |
| | | | | • Early sexual debut |
| | | | | • Insufficient and inaccurate knowledge about fertility and SRH services |
| | | | | • Beliefs about sexuality, sex considered taboo |
| **Zaman et al., 2012** [78] Asia and Pacific Islands | Current situation of fertility decline and status of FP programs in selected countries in Asia and the Pacific | Report | Not provided | • Initial FP were often embedded in economic development plans |
| | | | | • Some common characteristics across the Pacific but also great variation made it difficult to generalise. |
| | | | | • Analysis of Pacific programs showed a relationship between fertility transition and FP |
| | | | | • Some FP programs stalled, reversed or slowed down |
| | | | | • Adolescents faced largest barriers to the use of contraceptives for socio-cultural reasons |

## Results

After duplicates were removed, a total of 136 articles were identified from electronic databases and grey literature. Based on title and the abstract, 47 were excluded with 89 articles retrieved and assessed for eligibility. Of these, 44 were excluded for the following reasons: 16 studies did not focus on FP service provision; 13 papers were studies on SRH risks and behaviours; seven studies were from the DHS and MICS, which showed the same findings as the included DHS reports; four papers discussed reproductive health training and working frameworks and two papers were editorial and commentary. Another two studies were excluded because they were unable to be accessed. The remaining 45 papers, peer-reviewed (n = 15) and grey literature (n = 30), were reviewed in full (Table 1). Publication types were categorised according to the Sanson-Fisher typology (Table 1) [44]. The majority (80%) of these studies included Melanesian countries such as Solomon Islands, Fiji, Vanuatu and PNG and were mostly descriptive studies of limited quality, focused mainly on: knowledge, attitudes and barriers to FP; factors that influence use of contraception; and access to reproductive health services. Meaning units

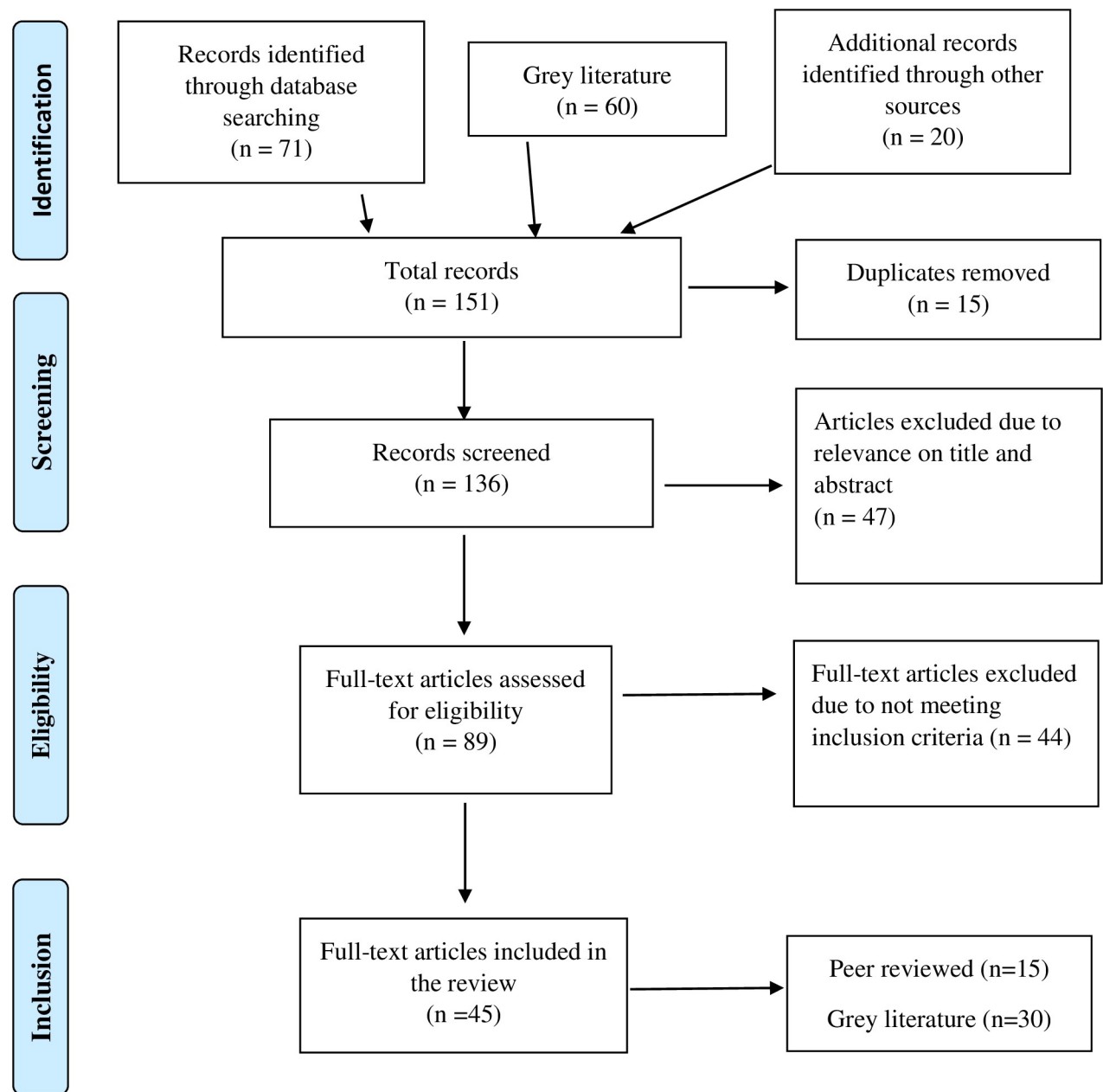

**Fig 1. PRISMA flow chart for inclusion of articles [43].**

from included papers were identified and extracted to develop codes, categories and themes (S6 Appendix) [39]. Five themes were identified from the literature, as outlined below.

## Family planning services in the Pacific

Family planning services in the Pacific region were available in most health facilities. Thirteen papers [5, 45, 50, 53, 58, 63, 66–69, 72, 76, 79] described government operated public health facilities as the main provider where contraceptives are usually free. A further five papers [5, 52, 53, 67, 72] described services provided in a variety of venues such as family planning associations, non government organisations (NGOs) and faith based organisations (FBOs). In

addition, FP services could also be obtained from private pharmacies and doctors working in private practices [50]. One paper reported people sourcing contraceptives from relatives overseas [53]. Overall, the most available way to access FP services was reported to be via the government public health system.

Pacific health facilities traditionally operated a once a week session for FP within the context of MCH/FP [34, 54]. This context now included RMNCAH, which means FP was made available along with other SRH services in the RMNCAH platform. Some settings also provided outreach RMNCAH to extend services to community villages [73]. Most health facilities in PICTs offered three to five modern contraceptive methods: oral contraceptive pills, long acting reversible contraceptives (injectable and implant), intrauterine contraceptive device, condoms and permanent methods (vasectomy and tubal ligation). Emergency contraceptives are the least known and used method and not available in most health facilities. The use of traditional methods such as the 'rhythm' and 'withdrawal' methods were also reported but were not recommended because they were unreliable [36, 50]. The availability of these methods depend on the level of health facilities. Higher-level facilities like hospitals and urban centres often provide comprehensive options compared to community facilities like rural health clinics and aid posts where less options and expertise are available [57, 74].

Overwhelmingly, 18 papers [2, 6, 30, 34, 46, 49, 54, 55, 60, 61, 63, 65–67, 72, 76, 78, 80] reported that FP services provided were not always accessible to everyone. Services that provide FP were perceived as inconvenient, unsatisfactory, and not culturally sensitive [51]. Services were usually female oriented; men were not involved and often not targeted in educational awareness of safe motherhood initiatives [54, 55, 78]. Married women aged 20 to 35 were the most common cohort accessing FP services [2, 46, 55, 63, 65, 66]. Unmarried women or unmarried adolescents (12–19 years) [2, 55, 61, 65, 67, 71, 78], older women (over 40 years) [30] and men aged 15–54 [47, 49, 54] years were relatively neglected in FP clinics. Pacific countries with dispersed island populations and remote locations also received poor services [63, 70, 72]. Although the emphasis on providing services for outer islands and rural communities has been articulated, the actual implementation has varied depending on available resources and support including staffing, political commitment and logistic systems. Evidence about the extent to which FP services are reaching those who use the service and how the service was provided at the health facility level was lacking in the literature [30, 47].

Most PICTs are committed to upholding human rights and have adopted policies based on the principle of free and 'informed choice' for all couples and individuals [50]. Reports on the existence and use of relevant SRHR policies in five Pacific countries (Kiribati, Samoa, Solomon Islands, Tonga and Vanuatu [70–74], including ICPD progress reports [3, 18, 21], showed some PICTs have used some form of SRH policies. Other PICTs have SRH policies in either draft form or under development. The incorporation of these policies into national strategies in PICTs occurred in varying degrees. These policies aim to provide an enabling environment where reproductive rights of women and men can be fully recognised and exercised. However, for those PICTs who have polices, most are outdated. In some cases, policies are not visibly available and approaches to FP shown to be ineffective [34]. For example in Tonga [73], there are no mentoring programs to monitor skills retention. In Kiribati [70] SRH care is not standardised across all health facilities due to lack of FP guidelines. Where guidelines and tools are available, they can only be effective if supplies to meet increased demand are available [59]. The UNFPA progress report [14] also highlighted that achievement of reproductive health rights are yet to be fully established in the Pacific region.

The integration of SRH, including FP within PHC service is part of the comprehensive SRH package prioritised in PICTs. Integration in this review refers to integration of population policies into sector plans and strategies and integration of FP services within the SRH platform

at the program and health facility levels. Hayes and Robertson [50] reported most Pacific countries have incorporated reproductive health including FP into national and subnational development plans. However, assessments of SRH programs in some Pacific countries showed integration was not fully implemented [70–74]. There is less progress in the integration of population policies into sector plans [21]. At the program level, although the MCH/FP focus has changed to RMNCAH to enhance inclusivity for everyone, there is little focus on STIs including HIV, abortion, infertility, and adolescent health by FP clinics [57, 72]. Evidence also suggests that clinics for STIs, including HIV, are delivered separately from routine reproductive health clinics in some PICTs [14, 28, 55, 62, 63, 65, 68, 69, 72].

The availability of quality and relevant SRH, including FP data was a common and ongoing challenge in PICTs. Robertson [36] and UNFPA [14, 70, 72, 74] documented poor reporting systems and processes that made it difficult to determine whether valid data existed for most countries. Prior to 2006, DHS were undertaken in only two countries in the region: PNG and Samoa. As such, the use of contraceptives could be under-reported. One example is the difference in CPRs reported in the DHS when compared to Ministry of Health (MOH) reports. This difference could be because women are accessing contraception from private pharmacies, private practitioner and other NGOs and this data is not routinely being captured in the national data [78]. Therefore, there is an urgent need to improve current reporting systems so that all parties communicate more effectively to produce an accurate reflection of CPRs in the Pacific context.

Information on unmet needs for contraceptives was not initially included [36]. The sociocultural and demographic diversity of PICTs made it difficult to interpret certain indicators and targets such as maternal mortality rates, within the context of very small populations. The disparities that existed across socio-economic groups contributed to the challenge of monitoring progress to achieve target goals [21]. While most countries are now able to conduct population census, the capacity to process, analyse and interpret survey results from policy perspectives remains limited [18]. Routine health information systems will need to be validated, and capture contextual issues, that can inform relevant policies and FP service outcomes.

## Education, knowledge and attitudes

The education, knowledge and attitudes of FP service users was a common theme among the reviewed literature [2, 5, 6, 46, 48, 49, 53, 54, 56, 58, 61–63, 65–69, 76, 77]. The service user's knowledge about the different contraceptive methods varied across countries by education levels, gender, marital status, parity, age group, wealth quintiles and where they live. While it is expected that women with higher education levels (more than secondary education) are more likely to use contraceptives to delay pregnancy because they may have greater exposure to contraceptive knowledge and options, some exceptions and inconsistencies were reported [46, 48]. For example, some women were unaware of important information about contraceptives, such as the availability and use of emergency contraceptives and the use of condoms to prevent pregnancy [61]. Nine reports of DHS conducted in PICTs between 2007 and 2018 [53, 58, 62, 63, 66–69, 76] showed a consistent result of almost universal contraceptive knowledge among married women and men above 40 years of age but lower knowledge levels in the unmarried and adolescent cohorts. Increased contraceptive knowledge was also seen in women with higher parity (>3) compared to women with lower parity (<2). Contraceptive knowledge according to wealth quintiles and rural or urban dwelling also showed inconsistent findings throughout PICTs. This knowledge is defined as having heard about a method, however, for adolescents, they lack knowledge about services and important information on pregnancy prevention, condom use, puberty, sexuality and relationships [2, 48].

Contraceptive use was relatively low in PICTs. Davis [49] and Naidu et al., [61] reported that having high contraceptive knowledge did not always translate to use of contraceptives and having heard about a method did not always influence individual decisions. For example, in Kiribati and Marshall Islands, contraceptive use was high among those with low or no education and lower among the highly educated populations, compared to Solomon Islands and PNG [53, 58, 63]. In Samoa, there was no difference by use in rural and urban settings [66], whereas in Tonga, women in urban settings are less likely to use contraceptives than their rural counterparts [68]. In PNG, men's educational background increases the likelihood of women's use of contraceptives, and male literacy contributed to men's participation in FP [54]. The DHS reports included in this review [53, 58, 62, 63, 66–69, 76] further detailed that more than half of the men interviewed say they knew that their wives or partners used contraceptives, but only a few countries reported it was a joint decision. In addition, many women in PICTs (>70%) did not intend to use contraception in the future regardless of their knowledge of contraceptives. Contraceptive knowledge, use and attitudes will need to be understood within the respective PICT contexts before relevant strategies can be implemented.

A health worker's education, knowledge and positive attitude is essential to the success of FP services in PICTs. Health workers need education to increase knowledge, confidence and skills to empower couples and individuals to make informed contraceptive choices [57]. Marshall [57] and the UNFPA report on the Pacific's RMNCAH workforce [81] reported that not all health workers had adequate knowledge about the risks, uses and options for contraceptives available to women. There was also a lack of knowledge and skills to dispense contraceptives and how best to deliver FP services. This includes the ability to give appropriate contraceptive counselling advice; practical skills to insert uterine devices or implants; and communicating with and managing adolescents [6, 48, 77]. This means planners in PICTs will need to explore if skill mix and task shifting will be beneficial in their context.

Some health workers also lack the training required and slowly abandon their moralistic attitudes to deliver effective adolescent SRH services [78]. In some PICTs contexts, even if training in FP was provided, negative attitudes and beliefs of health workers about contraceptives influenced whether the health workers promoted or discouraged adolescents from accessing the service and using contraceptives [5, 34, 60]. A recent report on the status of the RMNCAH workforce in the Pacific region [81] stated that most countries in the region have sufficient nurses who are competent to provide the RMNCAH care, including FP. However, there is also an overall shortage of nurse-midwives, and these healthcare workers will have multiple responsibilities additional to RMNCAH, meaning FP may not be prioritised or provided when needed. Adding to this challenge is the high staff turnover and the education and recruitment of RMNCAH workers, as many smaller countries did not have their own education institutions to provide this specialised training. For health workers working in remote areas, further training opportunities are often limited, therefore, the lack of training incentives also influenced the way health workers deliver FP services in PICTs.

### Geographical isolation and access

Accessibility of SRH services in PICTs is a particular challenge due to geography and climate. These countries are predominantly sparsely populated, small island nations dispersed across the Pacific. Some remote geographical locations make access to family planning services difficult [2, 73]. The majority of the population in most PICTs live in rural, often isolated areas or atolls with limited infrastructure such as roads, electricity and running water [14, 21, 55, 70, 73]. Access to health services is often threatened by bad weather, rough seas for those who live in coastal islands and flooded rivers for those who live on large islands [60]. People survive on subsistence farming

and fishing, and the health centre is often too far away to reach. Irregular supplies of medical equipment and drugs to the clinic can result in unavailability of contraceptives for women and men at point of need. The cost of travelling is expensive, and additionally prohibitive if people have to make return visits to the health centre or if contraceptives are out of stock [3, 9, 13, 21]. Access to FP services depends upon the adequate geographical spread of health facilities and a health workforce supported by reliable transport and communication networks [81]. Therefore, reliable and updated information about the country's health systems and effective planning are important to address resource allocations to prepare for expected and unexpected adverse situations.

## Socio-cultural beliefs, practices and influences

Most of the reviewed papers acknowledged the strong negative influence of socio-cultural and religious beliefs and practices relating to SRH issues in the Pacific [3, 5, 6, 14, 30, 34, 36, 46–49, 52, 54, 56, 58, 60, 62, 63, 65, 66, 68, 69, 73, 74, 76, 77]. Socio-cultural and religious beliefs and practices are very important to people in the Pacific. These beliefs and practices play a major role in community life, and the ubiquitous Christianisation of the Pacific by missionaries that enabled colonisation is associated with the idea of refusing contraception. [5, 60]. The common reasons reported in the literature as barriers to contraceptive use mainly rose from misconceptions, health concerns and a mixture of cultural and religious beliefs. Understanding and acknowledging these sociocultural influences is important to identify acceptable ways to reach people with FP services.

Although socio-cultural practices are seen to constrain progress in SRH including FP [3], in some PICTs, culture is viewed both as a way to promote, as well as constrain, SRH [21]. This reflects the hyper-diverse culture in the three main ethnic groups: Melanesia, Polynesia and Micronesia [14]. For example, in Samoa, women hold important traditional roles in society and can promote FP. In Kiribati, men have the traditional role in taking care of their wives during pregnancy and childbirth; therefore, they will have already fulfilled the role of men as partners in reproductive health [21, 70]. In Solomon Islands, an 'O clinic' (Ovulation clinic) was provided for those who wish to use the natural methods for cultural, religious or health reasons. Such opportunities to promote FP will need to be further explored when dealing with constraining issues in the socio-cultural context in PICTs [34].

The physical layout of health facilities and how services are delivered present barriers to service access in some PICTs. Burslem et al., [46] and Kennedy et al., [5] reported that some health facilities in Solomon Islands and Vanuatu, including those providing FP services, were considered not culturally suitable or accessible according to acceptable norms surrounding modern contraceptive use. These norms include gender-access issues, husband/partner opposition and beliefs that using contraceptives will encourage promiscuity, and use is therefore morally wrong for unmarried women and young people [34, 49]. One barrier that stands out in the reviewed literature is the culturally insensitive FP services that are not conducive for men or young people to access [36, 47, 49]. The gender of the service provider often affected men or women's access, and the lack of privacy and confidentiality was a common hindrance for young people and the marginalised groups [52]. While adolescents may have adequate knowledge about the importance of using contraception, they may be denied access because the service did not offer culturally appropriate options to ensure inclusivity [6, 46, 52, 55, 60].

The desire for many children is consistently described in the DHS reports, with men expressing wanting more children than women [53, 58, 62, 63, 66–69, 76]. Having larger families is an accepted cultural norm in the Pacific region. Children are valued as future social and economic gains and security [34, 50]. Although the number of children per woman in PICTs has been declining since the 1970s, one paper recently reported having three to five children

was seen as ideal [56]. It is important to note that remnants of traditional practices and ideologies to limit fertility still exist in some Pacific countries. However, Hayes and Robertson [50] considered using traditional 'modes of reproduction' to encourage Pacific Island people to adopt family planning has not been an effective strategy. The reasons are many and complex, and can be traced back to the initial contacts with missionaries and later colonial authorities, which resulted in the criminalisation of some traditional population control methods such as abortion and infanticide; postpartum abstinence and abstinence from sexual activity during ceremonial events. However, Lincoln et al., [56] suggested, if future family planning programs could also address the number of children in an ideal family unit, this could potentially be a way forward to ensure contraceptives are acceptable in PICTs.

With the introduction of pronatalist policies by missionaries and colonial authorities, some earlier control practices were considered immoral. Formal laws regarding marriage, births and deaths were formulated, and most derived from the Christian 'laws'. One belief the church has instilled, that has become a common religious barrier to contraceptive use, was that children are a "gift from God" and having more children is a good thing [50]. However, despite this barrier, some PICTs have found ways to deal with this belief. For example in Kiribati, although faced with religious opposition, the injectable contraceptive Depo Provera was acceptable and commonly used by Catholics, as opposed to longer-term methods which are considered unacceptable [53]. In Vanuatu, traditional leaders and religious groups were becoming more accepting of reproductive health and rights [21]. Hence, opportunities can be sought in the diverse cultures of PICTs to promote acceptable strategies to deal with religious beliefs.

## Potential enabling factors for improved family planning

Strategies to improve FP service provision and access in PICTs are outlined in nine of the reviewed papers [2, 36, 55, 59, 70–74].The importance of effective collaboration between government sectors, NGOs and private sectors was shown to increase access and avoid duplication of services. For example, adolescents in Vanuatu [5] prefer services provided by NGOs, as they were perceived to be more accessible, friendly and competent in helping people in this age group compared to government services. Integration of population policies such as SRH into other government sector plans were found to be beneficial in these small island states, where a lack of resources such as funding and staffing is prevalent [81]. Where contextual challenges such as cultural norms, religious obligations and geographic limitations occur, friendly service providers and context specific strategies are needed to implement relevant services [5, 47, 72].

Family planning education and health programs have been shown to positively influence contraceptive use. However, this review also identified that FP education in schools is underutilised in PICTs [2, 52, 57, 77] and there is a need to utilise peer educators, parents and schools to promote FP education in PICTs. Health education programs need to invest in a broad range of informational resources and utilise a multi-faceted approach to reach young people who are attending or not attending school and to reach other adolescents in both geographically isolated areas and urban settings [2, 52, 77]. In Tonga, a 'settings approach' to SRH program including awareness talks in schools, villages, workplaces, churches and radio talkback shows, was shown to improve young people's knowledge about SRH services [73]. Reproductive health and FP education have been implemented in PICTs, but education materials need to be translated into local languages and presented in culturally sensitive ways [28, 59]. Different strategies are required for male, female, adolescent or mixed audiences. Attaining higher education levels and obtaining education on SRH and FP are not enough [49]. Evidence-based motivational interviewing and behavioural change action are required to meet individual contraceptive needs and deal with contextual barriers.

Educating and involving men in reproductive health has been a missed opportunity to improve services [49]. It is important to engage boys and men early in the reproductive life cycle [49]. Most RMNCAH care services in PICTs do not actively engage expectant fathers and fathers of young children. An evaluation of male involvement in reproductive health (MIRH) in Fiji and Solomon Islands [64] and a study of MIRH in PNG [54] showed an acceptance of MIRH in Pacific contexts, with MIRH seen as a key opportunity to break down cultural barriers and norms to accessing FP services and the use of modern contraceptives by women and men. A qualitative study [49] on perspectives of policy makers and practitioners in Cook Islands, Fiji, PNG, Solomon Islands and Vanuatu also noted that the inception of MIRH prompted men to appreciate their role in FP and to better support their wives. However, context-appropriate strategies are needed for health service providers to engage men and deal with sensitivities relating to culture when discussing sexual health issues with them [36].

The increasing youth population and their early sexual debut in PICTs compared to other world regions suggest that this group warrants more attention [52]. Adolescent-focused services have been recently implemented but these lack culturally sensitive approaches, confidentiality and privacy [81]. In rural communities, engaging community gate-keepers in education awareness is vital [57, 73]. This will enable community involvement in distributing SRH information and extending FP awareness. Overall, FP needs further research to understand the changing behaviours of Pacific peoples so that SRH care including contraception service approaches are based on current evidence for PICTs [21, 45].

## Discussion

This is the first scoping review to explore and summarise the provision of FP services in PICTs. The results provide a baseline for researchers, policy makers and program managers as they seek to implement relevant strategies to improve FP services in the Pacific region. This review shows that PICTs did not follow the expectation of the standard demographic transition model. The transition from high to low fertility and death rates and increasing CPR did not translate to economic progress [78, 82]. Growing environmental pressures including urgent threats of climate change compound new challenges to population growth, increasing urbanisation and migration from rural to urban centres [83]. While some common characteristics are seen across PICTs in the provision of FP, progress and challenges, there are also great variations and diversities in country contexts, which make it difficult to generalise across countries. This is consistent with the WHO and UNFPA reports for LMICs [7, 29].

Since the ICPD, FP services in PICTs have been provided by government public health systems; while typically free of charge, FP services are not accessible to everyone. Subsequent reviews of ICPD progress [3, 18] revealed inequitable access to FP provided from health facilities. Men, adolescents and geographically marginalised groups including those on outer islands, are still not adequately reached [21]. The increasing growth in the youth population and a consistent lack of access to service among this group in PICTs warrants an urgent review of individual country strategies to reposition FP in country contexts.

At the political and policy levels, there is strong support and commitment for SRHR among PICTs and most have an official policy to guide SRHR implementation. However, these political commitments and policies have had little impact on achieving reproductive health and rights for PICT populations. This could reflect the ongoing global controversy about SRHR [84]. There are also possibilities that either research evidence has not informed current policies or health workers were not aware of existing policies. This is consistent with a study in PNG [85] where most health workers had not viewed official policy or statements about HIV.

Funding issues, contextual factors, staff shortages and the need for further training are ongoing challenges faced by PICTs and other LMICs [86, 87]. At the program level, many reproductive health programs have worked in "silos" (working in isolation, not sharing information, goals, priorities or processes) rather than integrating with other SRH services (for example STI services and FP) [72]. Existing strategies such as the RMNCAH to integrate family planning into other SRH services are not fully implemented [72]. This requires country-specific strategies to evaluate what may work best in each context. In addition, the inadequate reporting systems in PICTs raises the question of the quality of data that reports SRH and FP indictors. The targets or goals could be too ambitious for these small developing states to achieve. For example, patterns of unmet needs for FP and CPRs varied so much across PICTs that it was necessary to take a country-by-country approach [88].

Health workers often lack appropriate knowledge about FP services [14]. Staff numbers, skills and resources including funding remain inadequate [89]. Strategies such as skill mix and task shifting have been found to alleviate workforce challenges and skill mix imbalances in low income countries [90]. It is essential to engage and collaborate with local community leaders and women's groups to empower the community, as they know what is required to inform strategic planning to enhance universal access. One program in Fiji demonstrated the power of moving out of silos in health settings and engaging community members [91]. This empowerment program including workshops attended by disempowered young mothers was conducted in a rural community with high rates of teenage pregnancies and low contraceptive use. Topics such as reproductive health and rights, available support services, networking and financial literacy were presented. The results revealed a 30% increase in uptake of SRH services and young mothers were motivated to make positive changes in their lives [91].

The PICTs are culturally and spiritually diverse, and this needs to be taken into account when planning and delivering health services. This diversity is reflected in the way men participate in FP and how health workers deliver services. In most PICTs, the negative attitudes of some health workers and service users towards FP and modern contraception have widened the gap between knowledge and practice [6, 61, 77]. These attitudes are mainly influenced by the socio-cultural and religious beliefs of people. A study on community influences on young women in LMICs concluded that young women's contraceptive decision-making is greatly shaped by their social contexts [92].

However, cultural shifts in societal attitudes observed in some PICTs may facilitate progress [88, 93]. In Vanuatu, village leaders are now more open to discussions about SRH and young people are more receptive to information, so this can be a potential way to explore how to deal with cultural and spiritual barriers to access [21]. Culture and religion influence access to and use of contraception and could serve as barriers. Unexpectedly, we found PICTs have experienced situations when religion and culture could support a process towards FP [78]. For instance, the Catholic Church policy may not promote the use of modern contraception but international evidence suggests that people who identify as Catholic do make use of contraceptives [94, 95]. Although patterns of unmet need in contraception varied among PICTs, analysis from DHS reports showed that the main reason for high unmet need was not access but 'unwillingness' arising from fear of side-effects, health concerns and some form of socio-cultural opposition [10, 78]. This means DHS reports may need further analysis.

The relationship between education and contraceptive use is inconclusive in PICTs. As evidenced in the literature, despite several decades of FP programs including Information, Education and Communication (IEC) campaigns to improve knowledge and awareness of contraceptive methods, women continue to report lack of knowledge and fear of side effects [60, 61]. In addition to providing relevant and simplified IEC materials to increase understanding, spousal communication and male involvement in decision–making can positively influence FP use and continuation [96]

Education alone is not enough; dealing with barriers in a culturally sensitive manner may reduce socio-cultural issues. A study based in Timor-Leste and PNG revealed that although men had good knowledge in some areas of SRH, their attitudes regarding gender roles and violence against women reflected the social norms of more patriarchal societies [97]. In these settings, most men view the husband as the primary decision-maker, and a small percentage of men surveyed believed it was acceptable for a husband to beat his wife. However, in another study conducted in rural PNG, young men were more receptive to biomedical information than older men, and were more likely to engage with health services directly and support their wives to use implants [98]. In PICTs, men's involvement showed improvement in women's access to FP services, but culturally appropriate strategies are needed to ensure universal access by men. Achieving gender equality in SRH will require strategies to be explored for their applicability and sustainability in local settings.

The ways that health services are delivered and the location of FP clinics are causes for limited access. A recent study from a setting that holds strong cultural taboos in reproductive issues in Solomon Islands showed that the FP clinic could not be accessed because of its location and how it was made available [99]. Inappropriate models of service delivery were identified in this review, such as service providers' insensitivities to different cultural, social and gendered groups. Furthermore, evidence shows that adolescents do not want a separate clinic, only a friendly service provider who is non-judgmental and assures confidentiality [5, 48, 52, 57, 61, 100]. A review of confidentiality in FP services for young people by Brittain et al., [101] concluded that further research should consider how to best educate young people and providers about state-specific laws related to adolescents and confidential healthcare services, as there is limited research on the relationship of confidentiality and reproductive health outcomes in young people.

The PICTs are culturally and geographically diverse and one approach does not fit all, but there is potential for change at the health service level and for contextual approaches to FP to improve service and access to contraceptives. Given that women in PICTs have more options of modern contraceptives than men, further studies are needed on strategies to enable men in PICTs to fully engage in decision-making regarding the number of children, to ensure universal access.

The following recommendations are based on the evidence presented in this review:

1. The rights-based approach to SRH, including FP, as outlined in the ICPD program of action [29], needs to be culturally contextualised in PICTs.

2. Current approaches to service delivery need to reflect the reproductive and contraceptive needs of users and potential users of the service.

3. Appropriate and relevant community engagement, education and awareness tailored to meet community needs is required.

## Limitations

We found limited peer-reviewed studies conducted on FP service provision in most PICTs, therefore the results in this review may not represent and reflect issues for each PICT. The scoping review methodology may not have identified all sources. Results as reported could be limited by the methodological quality of the articles, as the majority of papers included are from the grey literature. The quality of data analysed in the papers may be of low quality given the challenges of poor reporting systems and incomplete data available in PICTs. Hence, reported literature may not give an accurate and clear picture of the situation in all PICTs.

## Conclusion

Family planning services in PICTs do not reach many people: a person's age, gender, marital and social status, religion, ability or proximity to health centres can impact access to FP services. If this trend continues, universal access to reproductive health services including FP will be an ongoing challenge. The higher education level of girls, high contraceptive knowledge and mostly free services in PICTs do not necessarily lead to increased use of modern contraceptives and access to FP services. Many contextual challenges remain in each Pacific country and territory in terms of both supply and demand side of FP and SRH services. Considering the heterogeneous cultural diversity of PICTs, generation of local evidence is crucial to make relevant and sustainable improvements in service delivery. Further research is required to understand availability, accessibility and acceptability of current FP services to meet the needs of people of different genders, age groups, and social and marital status to inform FP services in PICTs that leaves no one behind.

## Supporting information

**S1 Appendix. Review protocol.**
(PDF)

**S2 Appendix. PRISMA-ScR checklist.**
(PDF)

**S3 Appendix. Eligibility criteria.**
(PDF)

**S4 Appendix. Key words.**
(PDF)

**S5 Appendix. Search strategy.**
(PDF)

**S6 Appendix. Selection of sources of evidence.**
(PDF)

**S7 Appendix. Data charting.**
(PDF)

**S8 Appendix. Example coding process.**
(PDF)

## Acknowledgments

The authors would like to thank James Cook University Librarians, for assistance with database literature searches and Endnote software referencing and Dr Karen Cheer for editing the manuscript.

## Author Contributions

**Conceptualization:** Relmah Baritama Harrington, Nichole Harvey, Sarah Larkins, Michelle Redman-MacLaren.

**Data curation:** Relmah Baritama Harrington, Nichole Harvey, Sarah Larkins, Michelle Redman-MacLaren.

**Formal analysis:** Relmah Baritama Harrington, Nichole Harvey, Sarah Larkins, Michelle Redman-MacLaren.

**Investigation:** Relmah Baritama Harrington, Michelle Redman-MacLaren.

**Methodology:** Relmah Baritama Harrington, Nichole Harvey, Sarah Larkins, Michelle Redman-MacLaren.

**Project administration:** Relmah Baritama Harrington, Nichole Harvey, Michelle Redman-MacLaren.

**Resources:** Relmah Baritama Harrington, Nichole Harvey, Sarah Larkins, Michelle Redman-MacLaren.

**Software:** Relmah Baritama Harrington, Michelle Redman-MacLaren.

**Supervision:** Sarah Larkins, Michelle Redman-MacLaren.

**Validation:** Relmah Baritama Harrington, Nichole Harvey, Sarah Larkins, Michelle Redman-MacLaren.

**Visualization:** Relmah Baritama Harrington, Nichole Harvey, Sarah Larkins, Michelle Redman-MacLaren.

**Writing – original draft:** Relmah Baritama Harrington, Michelle Redman-MacLaren.

**Writing – review & editing:** Relmah Baritama Harrington, Nichole Harvey, Sarah Larkins, Michelle Redman-MacLaren.

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
