## [Decision Letter · Decision Letter 0]

18 Nov 2020

PONE-D-20-25718

Family planning in Pacific Island Countries and Territories (PICTs): A scoping review

PLOS ONE

Dear Dr. Harrington,

Thank you for submitting your manuscript to PLOS ONE. After careful consideration, we feel that it has merit but does not fully meet PLOS ONE’s publication criteria as it currently stands. Therefore, we invite you to submit a revised version of the manuscript that addresses the points raised during the review process.

The article is a scoping review, which PLOS ONE considers to be in scope as a type of systematic review and/or meta-analysis. Our author guidelines for systematic reviews/meta-analyses are at http://journals.plos.org/plosone/s/submission-guidelines#loc-systematic-reviews-and-meta-analyses. The required PRISMA-ScR checklist is provided (S2 table) as well as the flowchart (figure 1).

There are concerns raised by the reviewers on the selection of sources. It seems that the review is missing an important part of the literature both from peer-reviewed references and from the grey literature. These concerns should be addressed in order to provide a balanced view of what is known regarding FP in the pacific.

In particular, it seems that the search has not been as systematic as desirable regarding grey literature. There are evident places where it is necessary to look that have not been inspected. In particular, the United Nations Population Division publishes every two years the database on World Contraceptive Use (the last one from 2020), that includes sources. The contents of this database provide the primary evidence for monitoring of family planning SDGs. For each of these sources there generally is a survey report. Many are missing, as pointed out by reviewer 2 regarding MICS/DHS surveys. The same applies to work connected with FP from international organizations as commented by reviewer 2, including UNFPA needs assesments which should be available from UNFPA regional websites, and HIV/AIDs surveillance studies and youth risk behavior surveys from the region (reviewer 2). It seems the protocol was not working in this case. In this respect the pacific data hub https://pacificdata.org/ , SPCs Central Data Catalog, https://microdata.pacificdata.org/index.php/catalog that generally includes the survey reports, in addition to UNFPA and the UN Population Division are places that must be considered.Reviewer 1 identified loopholes in the protocol that lead to relevant studies from peer reviewed sources not being included, including not looking for the individual country names separately. These should also be addressed.In order to better place the study in context, as suggested by reviewer 3, a small summary regarding contraceptive prevalence and unmet need for family planning in the region compared to other relevant regions based on international data would enrich the study.There are also comments and suggestions by reviewers 1 and 3 regarding the interpretation of results, in particular in connection with the role of religion.

In any case, it is felt that a major write-up would be required since a more inclusive analysis of published reference might well lead to different conclusions. That is the first step that needs to be done in order to make the results of the analysis trustworthy.

We look forward to receiving your revised manuscript.

Kind regards,

José Antonio Ortega, Ph.D.

Academic Editor

PLOS ONE

Journal Requirements:

2. Please update the literature search to include publications since 2018.

Reviewers' comments:

Reviewer's Responses to Questions

**Comments to the Author**

1. Is the manuscript technically sound, and do the data support the conclusions?

Reviewer #1: Yes

Reviewer #2: Yes

Reviewer #3: Yes

2. Has the statistical analysis been performed appropriately and rigorously? 

Reviewer #1: N/A

Reviewer #2: N/A

Reviewer #3: N/A

3. Have the authors made all data underlying the findings in their manuscript fully available?

Reviewer #1: No

Reviewer #2: Yes

Reviewer #3: Yes

4. Is the manuscript presented in an intelligible fashion and written in standard English?

Reviewer #1: Yes

Reviewer #2: Yes

Reviewer #3: Yes

5. Review Comments to the Author

Reviewer #1: Abstract

Authors highlighted the role of “culture and religion” as the main barriers of accessing family health services. It was extracted form the reference 36 and 42 while these two main findings are not presented in the data extraction sheet. Moreover, it is very difficult to accept, in Pacific country. Culture and religious always play the role as barrier.

Introduction

1. Overall, this section mostly focuses on history of family planning instead of the importance, challenges and actually the gap which leaded to conduct this research.

1- Page 3, line 2, please remove the dot (.) before the reference 1 in page 3.

2- Page 3, line 7, it is needed to use (SRH) after the “sexual and reproductive health” as the abbreviation is used later.

3- Page 4, last line, you can only use “SRH” instead of the full words.

4- Page 6, line 4, you can only use “SRH” instead of using its complete words.

Method:

1- Overall, this section lacks a good search strategy, appraisal process, and extracting data.

2- Page 6, the last 3 lines, you can use “STH” instead of suing its complete words.

3- Page 6, the last 2 line, you can use “PICs” after “Pacific Island Countries”.

4- Page 7, number 4, it is not clearly mentioned the period/time of searching paper. What does “Current” mean in the sentence “Published between 1994 and current”?

5- Page 7, line 6, you can use “SRC” instead of “sexual reproductive health”.

6- Page 7, lines 4 and 5 of “Information sources and search”: why you used * after melanesia* polynesia* micronesia*? This is not correct usage of (*).

7- Page 7, lines 4 and 5 of “Information sources and search”: it is needed to use “OR” between “Melanesia, Polynesia, micronesia”.

8- Using a keywords of “pacific island countries and territories” is a keywords that caused authors to lose many studies. It can be searched even using “Pacific” or the names of each Pacific countries (as listed in page 5 of “Introduction” to allow authors to find relevant studies. For example, I know there are more than 5 studies that are published by this reviewer while the author lost in this search. They are available in google scholar as the authors mentioned they used it for their search.

9- Page 7, lines 4 and 5 of “Information sources and search”: it is not clear why the authors used (*) after “challenge*, success*, barrier, enabler”? Using (*) is only to find the words that are similar such as “Prevention OR preventive”. In this case they could use “Prevnti*”.

10- Page 7, Lines 4 and 5 of “Information sources and search”: authors could use “OR” to include more synonyms and allow the databases to find more studies. For example it should be searched like “barrier OR Challenge OR obstacle”…. This caused authors to lose many studies.

11- Same comments for “wom?n OR m?n OR “young people”. That is one of the reasons that authors lost many relevant studies.

11- Page 7, second paragraph of “Information sources and search”: it is not clear why and how the authors chose “SCOPUS, MEDLINE and CINAHL” as databases for the search? There are other relevant databases that could help them to reach more relevant studies like CINAHL, PsycInfo.

12- Page 8, line 6, it is better to mention “data extraction sheet” instead of “Key findings” as this table presents all the information extracted for the selected studies.

13- Page 9, please remove “missing full text” from the box that included 73 studies. This stage authors just focuses on the title and abstract NOT the full text.

14- Page 9, last box where the “Inclusion” is achieved, the types of studies should be presented instead of writing only 22.

15- This section there is no any information about studies appraisal. If authors used PRISMA checklist, they could assess the quality of studies to include high quality studies in the analysis.

16- The main issue in this section is authors didn’t do bibliography search of remained 22 studies (after the last stage). That is why they lost many studies that may not published in the selected databases and other sources but they could easily find them in the bibliography.

Results:

1- Overall, this section is written well but there are many results that are extracted from the 22 studies that were included in the final analysis. This is the big issue.

2- Page 10, line 2 of “results”: authors written “Based on title and the abstract, 73 were excluded”. This has been presented incorrectly in PRISMA flow chart present before that. They believe they considered the full text too.

3- I strongly believe, authors lost many studies that could be included in this study. It is not correct that only 12 original studies are conducted on family planning in Pacific. There is a big gap between the found studies and real available studies.

4-. Page 11, Table 4, the tile should be “data extraction sheet”.

5- There are many information extracted from the studies that are not presented in Table 1 while they are presented in the finding section. For example “culture and religion” was considered as the main berries for theme “Enabling factors”. The reference 32 and 46 were authors got this findings while there is no any information about the barriers from these two references in Table 1. There are many other similar errors that shows authors extracted results while these key findings are not presented in the Table 1.

6- Page 18, line 2 of “Geographical isolation and access” the results extracted is form the reference 6 while this reference is not belong the 22 studies summarized in Table 1.

7- Page 18, line 10, same wrong issues has happened. There is no any reference 6 and 12 as the studies that were summarized in Table 1 while the results extracted form these references.

8- Page 18, line 2 of “Socio-cultural beliefs and influences” there is results extracted form reference 49 while this reference is not part of 22 studies that summarized in Table 1.

Discussion:

1- This section is written very poor. There is no good justifications for many results presented in the results section.

2- Page 21, line 11, authors are using SRH while they could use this abbreviation earlier as I mentioned in my previous comments.

Limitations:

There are lots of limitations that authors didn’t mentioned in this section.

Reviewer #2: Congratulations to the authors on a solid research paper and shining the light on family planning services and uptake in the Pacific region as there is not enough research about these broader dimensions of sexual and reproductive health and rights. The conclusions are valid based on the 22 papers reviewed. I would like to qualify this feedback in the context that this is the first time I have participated in a review such as this and I hope I have followed the guidelines correctly.

In terms of suggestions of how the paper might be improved and references to human rights, perhaps the scope of the (normative) frameworks should encompass not only ICPD, MDGs and SDGs; but also the Beijing Declaration and Platform for Action, as well as specific reference to CEDAW, CRC and CRPD etc. to strengthen the reference to human rights. The introduction could potentially include references to public/private SRH providers and increasing use of emergency contraceptives, traditional family planning practices, increasing use of the private sector because of perceived limitations through state clinics, the evolution of SRH in education curriculums and the ‘family life’ model, drawing from the Demographic and Health Surveys ‘empowerment’ findings and teasing out women’s limited autonomy over decision making to name but a few. However, if as it seems, this information wasn’t gleaned from the search results, then perhaps some more general references to these broader issues could be incorporated in the narrative.

However, I am somewhat perplexed and surprised by the overall search results, noting the rigorous method and criteria employed. For example, I would have thought that the search would have returned papers either referring to or specifically about:

• Second generation surveillance for HIV/AIDs studies conducted throughout the region that focused on youth and SRH etc;

• Demographic and Health Survey reports from almost all PICs governments, with the search only finding the Solomon Islands DHS report (Please note the correct survey name for page 7); and

• UNFPA SRH Needs Assessments Health Facility Readiness Assessments as well as the RMNCAH Workforce Assessment – for a number of PICs as well as the Solomon Islands.

It would be good if references to the dates of the publications ‘published between 1994 and current’, ‘published between 1994 and 2019’ used in the paper, are consistent and more precise in terms of ‘published between 1994 and <<month>> 2019’.

As noted, I stand ready to revise this based on feedback from the editor.</month>

Reviewer #3: On the overall, the conclusions the author has made are accurate. It would be good to note, that not all of the Pacific Island countries are equally in their development status, and a history of conflict in countries like PNG and the Solomon Islands which delay progress towards development. In the early part of the paper, it may be useful to utilise data on the gaps ie unmet need and contraceptive prevalence rates in the PICTs to illustrate the lack of uptake, and hence the need for this paper. This data is available through reports such as the Demographic Health Surveys, and the World Contraceptive Use Reports by the UN Population Division. This can help show the differences between the PICTs, and other countries, as well as progress made over time.

The effect of culture and religion in the Pacific islands on issues of sexuality and reproduction, and women’s rights cannot be underestimated. In fact two of the seven countries – who have not signed or ratified CEDAW come from the PICTs – Tonga (not signed, not ratified) and Palau (signed, not ratified) along with countries like the Holy See, the US, Iran, Sudan and Somalia. Additionally, in the introduction, the authors write that in response to China’s practices of coercive contraception and abortion, the Holy See issued a strong stance to these issues - (In response to the controversy, the Roman Catholic Church prohibited all artificial contraception and made a strong stand against abortion as a method of family planning [11].)

This is inaccurate as 1) the Holy See has had centuries-long tussle on the issue of abortion (beliefs ranging from early abortion ie before ensoulment was not murder, until 1869 all forms were outlawed) and contraception prior to China’s actions. 2) The Catholic Church’s stance against abortion is not only as a “method of family planning” but for even therapeutic reasons – rape, birth defects.

An older study – but still useful to consider is Donald P Warwick’s 1988 “Culture and the Management of Family Planning Programs” published by Population Council, which brings together key considerations which include aspects such as kinship ties, value of children, communal divisions which help advance or hold back success of contraceptive programmes. Some of these factors may also be at play in the PICTs, rather than just ‘religious resistance.’

6. PLOS authors have the option to publish the peer review history of their article (what does this mean?). If published, this will include your full peer review and any attached files.

Reviewer #1: No

Reviewer #2: **Yes: **Kim Robertson, Gender Statistics Advisor, SPC

Reviewer #3: No

---

## [Author Response · Author response to Decision Letter 0]

6 Jun 2021

Dear Editor,

Thank you for the opportunity to respond to reviewer comments. Please see authors’ responses below

Reviewer’s comments / Changes made /Author’s comments

Academic editor

1. In particular, it seems that the search has not been as systematic as desirable regarding grey literature. There are evident places where it is necessary to look that have not been inspected… UN population Division on world Contraceptive Use, MICs/DHS surveys, UNFPA needs assessments, HIV/AIDS surveillance studies and youth risk behaviors. Places must be considered are SPCs Central Data Catalog, UNFPA and UN Population Division The search strategy is revised to include all relevant places recommended and additional grey literature including DHS, relevant UNFPA reports are included This is addressed in response to reviewer 1 comments

2. Reviewer 1 identified loopholes in the protocol that lead to relevant studies from peer reviewed sources not being included, including not looking for the individual country names separately. These should also be addressed The protocol was revised and individual country names are searched. The changes made to this comment is addressed below under the section that responded to reviewer 1 comments

3. In order to better place the study in context, as suggested by reviewer 3, a small summary regarding contraceptive prevalence and unmet need for family planning in the region compared to other relevant regions based on international data would enrich the study A summary of contraceptive prevalence rates and unmet need for family planning were included in the introduction This is also addressed below in response to reviewer 1 comments

4. There are also comments and suggestions by reviewers 1 and 3 regarding the interpretation of results, in particular in connection with the role of religion This information is corrected and further gleaned from the evidence identified in the included papers 

Reviewer 1

Introduction 

Overall, this section mostly focuses on history of family planning instead of the importance, challenges and actually the gap which leaded to conduct this research “Introduction”

p. 2-6 , major revision done to this section to present the importance, challenges and gap Thank you for your feedback. Part of the history discussing family planning has been removed. The importance and challenges were incorporated to present a clear picture of the gap for this review. 

1. Page 3, line 2, please remove the dot (.) before the reference 1 in page 3 P. 2, paragraph 1, line 2.

… pregnancy and childbirth complications [1-3]. Dot removed.

2. Page 3, line 7, it is needed to use (SRH) after the “sexual and reproductive health” as the abbreviation is used later P. 4, paragraph 1, line 4. 

…sexual and reproductive health (SRH) services, … “SRH” inserted after sexual and reproductive health.

3. Page 4, last line, you can only use “SRH” instead of the full words P. 5, paragraph 1, line 2. 

 “SRH” inserted, full words deleted.

4. Page 5, paragraph 2, line 6 P 5, paragraph 2, line 14 “adolescence” included Change made

5. Page 6, line 4, you can only use “SRH” instead of using its complete words P. 5, paragraph 2, line 12. “SRH” used “SRH” used instead of full words.

Method

1. Overall this section lacks a good search strategy, appraisal process and extracting data “Method section” p.6, second paragraph to page 10, line 2. “search strategy” revised and a new search was executed Search terms, key words and subject heading were revised to include all relevant steps and a whole new search was conducted on the databases and relevant websites

2. Page 6, the last 3 lines you can use “STH” instead of using its complete words Page.6, paragraph 1, line 9. Full words deleted, “SRH” used in text I don’t know what you mean by “STH”, but I assume it’s a typing error so I made the changes to “SRH” instead

3. Page 6, the last line you can use “PICs” after Pacific Island Countries Page.6, under “Eligibility criteria”, no 1, last line. I changed “pacific island countries” to “countries in the pacific” To be consistent throughout, I did not use the “PICs” in this scoping review

4. Page 7, number 4, it is not clearly mentioned that the period/time of searching paper. What does “Current” mean in the sentence “Published between 1994 and current Page. 7, number 4, first line, I changed the “Current” to “2019” I meant this to be 2019, the time when I did the search for this scoping review, so I changed to 2019.

5. Page 7, line 6, you can use “SRC” instead of “sexual reproductive health” Page.7, line 5, “Papers discussing “ sexual reproductive health…” was changed to “Papers discussing SRH…” I did not know what the reviewer meant by “SRC”. I assumed it was a typing error that was supposed to be “SRH”

6. Page 7, lines 4 and 5 of “information sources and search”, why you used *after Melanesia, Polynesia* Micronesia? This is not correct use of (*) p.7, under “information sources and search”. I removed the key words and search terms from the manuscript, under the “Information sources and search” and included them in the search strategy as an appendix In databases including MEDLINE, the truncation symbol (*) can be used to find variant word endings. I used the * as truncation in my search for Melanesia, Polynesia and Micronesia to retrieve variant endings of these words.

7. Page 7, lines 4 and 5 of “information sources and search”: it is needed to use “OR” between “Melanesia, Polynesia, Micronesia Page. 7, line 4 and 5, under “information sources and search”. The search strategy corrected and removed from the manuscript. The full search strategy can be found in the appendix (S4 Appendix) Search strategy revised and corrected and inserted in the “search strategy appendix”

8. Using a key word of “pacific island countries and territories” is a key word that caused the authors to lose many studies. It can be even searched using “Pacific” or the names of each Pacific countries (as listed in page 5 of “introduction to allow authors to find relevant studies. For example I know there are more than 5 studies that are published by this reviewer while the author lost in this search. They are available in google scholar as the authors mentioned they use it for their search Under “information sources and search”, page 7, the key words: “pacific”, “pacific islands”, pacific island countries”, “pacific region”, pacific island countries and territories”, “american samoa”, “cook islands”, “federated state of micronesia”, Fiji”, “french polynesia”, “guam”, “kiribati”, “marshall islands”, “nauru”, “new caledonia”, “niue”, “northern mariana islands”, “palau”, “papua new guinea”, “samoa”, “solomon islands”, “tokelau”, “tonga”, “tuvalu”, “vanuatu”, “wallis and futuna” were used to repeat the search again on databases and google scholar The key word was executed as per the authors changes and each country in PICTs were searched individually (see search strategy appendix)

9. Page 7, lines 4 and 5 of “information sources and search”: it is not clear why the authors used (*) after challenge*, success* barrier and enabler? Using (*) is only to find the words that are similar such as “Prevention” OR “preventive”. In this case they could use “Preventi*” page. 7, under “information sources and search”. As in comment 6 above, the same change was made here as well. As in comment 6 above, the * was used for truncation for key words as “challenge*” for “challenges”, “success*” for “successes” or “successful”, “barrier*” for “barriers”, “enabler*” for “enablers” 

10. Page 7, lines 4 and 5 of “information sources and search”: authors could use “OR” to include more synonyms and allow the databases to find more studies. For example it should be searched like “barrier OR “Challenge” OR “obstacle”… This caused authors to lose many studies. Page7, under “information sources and search”, “OR” has been used between the synonyms to include more sources from data base searches. The same as done for comment 7 above. The search was done again to include synonyms with “OR”

11. Same comments for wom?n OR m?n OR “young people”. That is one of the reasons that authors lost many relevant studies Under “information and search”, page 7, search terms for “wom?n” and “m?n” were removed from the manuscript and put in the search strategy as an appendix The “?” is used here as a wildcard to search for alternative words, for example “wom?n” retrieves both “women” and “woman” 

11. Page 7 second paragraph of “information sources and search”: it is not clear why and how the authors chose “SCOPUS, MEDLINE and CINAHL” as databases for the search? There are other relevant data bases that could help them to reach more relevant studies like CINAHL, PsychInfo. Page 7, under “information sources and search”, the reason for authors choice of “SCOPUS”, “MEDLINE” and “CINAHL” and “PsycINFO” now included as follows: “These databases were chosen as they provide most relevant peer reviewed articles about family planning in the Pacific”. PsycINFO database was searched but omitted from the text. It is now included. Explanation of database selection inserted 

12. Page 8, line 6, it is better to mention “data extraction sheet” instead of “Key findings” as this table presents all the information extracted for the selected studies page. 8, line 2, under “selection of sources of evidence”, the term “key findings” is deleted and replaced with “data extraction sheet” as label of table “findings” removed and “data extraction sheet” inserted

13. Page 9, please remove “missing full text” from the box that included 73 studies. This stage author focuses on the Title and abstract NOT the full text Page.9, “PRISMA flow diagram”, Figure 1,.The phrase “missing full text” was removed from PRISMA diagram box on “articles excluded due to relevance on title and abstract” The number of articles in the PRISMA diagram corrected and updated

14. Page 9, last box where the “inclusion” is achieved, the types of studies should be presented instead writing only 22. Page.9, last box where inclusion is achieved. A new box is inserted on the right, with the number of peer reviewed and grey literature inserted The number and type of literature were inserted in the box

15. This section there is no any information about studies appraisal. If authors used PRISMA checklist, they could assess the quality of studies to include high quality studies in the analysis Page 9, last line, under “data charting process”, A statement about quality appraisal of selected studies included as:… “Given that limited peer reviewed research conducted on family planning service provision in the Pacific region, studies…” Given the paucity of literature in this space, a scoping review methodology was employed to capture literature of varying quality. A comment about methodological quality of included studies has been made in the Discussion

16. The main issue in this section is authors didn’t do bibliography search of remained 22 studies (after the last stage). That is why they lost many studies that may not published in the selected databases and other sources but they could easily find them in the bibliography A bibliography search was done to search for relevant articles A bibliography search was done to look for relevant papers and this has been clarified. 

Results

1. Overall, this section is written well but there are many results that are extracted from the 22 studies that “were included in the final analysis”. This is the big issue “results”, page. 10, paragraph 2 to page 34, line 14. Papers were reviewed again to ensure information related to the research questions were extracted and included in the final analysis. Information not related to review questions were removed. Not sure if the reviewer meant to say “were not included”?

2. Page 10, line 2 of “results”: authors written “Based on the title and abstract, 73 were excluded”. This has been presented incorrectly in PRISMA flow chart present before that. They believe they considered the full text too. Page. 10, under “results” line 2. This was corrected and updated with results of the new search The error was corrected and relevant information updated

3. I strongly believe, authors lost many studies that could be included in this study. It is not correct that only 12 original studies are conducted on family planning in Pacific. There is a big gap between found studies and real available studies. Page 10, under “Results”, line 9, references to the Sanson-Fisher typology used was included The description of research types in this review has been categorized according to the Sanson-Fisher typology.(2006) “Original research” refers to studies that are data based. This can be data or new analysis of existing data relating to health issues for indigenous people. (Sanson-Fisher RW, Campbell EM, Perkins JJ, Blunden SV, Davis BB. Indigenous health research: a critical review of outputs over time. Medical Journal of Australia. 2006;184(10):502-5.)

4. Page 11, table 4, the title should be “data extraction sheet” Page. 11-24, table 1 “Table 1 Title: Summary of selected papers” was changed to “Data extraction sheet” The title of “Table 1” was changed. I guess “table 4” written by the reviewer may refer to “table 1 here”

5. There are many information extracted from the studies that are not presented in table 1 while they are presented in the finding section. For example “culture and religion” was considered as the main barriers for theme “Enabling factors”. The reference 32 and 46 were authors got this findings while there is no any information about the barriers from these two references in Table 1. There are many other similar errors that shows authors extracted results while these key findings are not presented in table 1. Page.10-34, paragraph 1 and 2. As in the “Results” section no: 1 above. Papers were reviewed again to ensure information related to the research questions were extracted and included in the final analysis. The references were also checked and corrected. Errors were checked and corrected

6. Page 18, line 2 of “Geographical isolation and access” the results extracted is from the reference 6 while this reference is not belong to the 22 studies summarised in Table 1 Page.30, line 3, under “geographical isolation and access”. The incorrect “reference” checked and removed. Correct reference inserted Correction was done

7. Page 18, line 10, same wrong issues has happened. There is no any reference 6 and 12 as the studies that were summarised in Table 1 while the results extracted from these references Page. 30, line 11, under “geographical isolation and access”. References checked again. Correct reference inserted Correction done

8. Page 18, line 2 of “socio-cultural beliefs and influences” there are results extracted from reference 49, while this reference is not part of the 22 studies that summarised in Table 1 Page. 30, line 2-3, under “socio-cultural beliefs, practices and influences”. The reference checked and corrected. Correction done

Discussion

1. This section is written very poor. There is no good justifications for many results presented in the results section Page. 34-38, last paragraph, under “discussion” This section was revised again and updated with the results section and re-written The issue is addressed during the revision

2. Page 21, line 11, authors are using SRH while they could use this abbreviation earlier as I mentioned in my previous comments Page.35, paragraph 2, line 12. The term “SRH” was checked again and abbreviation is used for consistency Addressed during revision

Limitations

There are lots of limitations that authors did not mention in this section

 Page. 38, under “limitations” Additional limitations of the review were added Addressed during revision

Reviewer 2 

Introduction

The introduction could potentially include references to: 

1. Public/private health providers This is included in the results This is addressed in the revision

2. Increasing use of emergency contraceptives P. 25, para 2, line 7 under “Family planning services in the Pacific” : references to the use of emergency contraceptives were made in the results section I did not find clear evidence about increasing use of emergency contraceptives in the Pacific from the literature. The evidence from reports and Demographic and Health Surveys showed: emergency contraceptives are the least known and used method, among Pacific women. Not generally available in most health facilities

3. Traditional family planning practices References to “Traditional family planning practices” were included in the “results” section, under “socio-cultural beliefs, practices and influences”, page 32, paragraph 1. References were made to traditional family planning practices during the revision

4. Increasing use of private sector because of perceived limitations through state clinics There is evidence of preferences for private sector over state clinics as they are perceived to offer better and friendlier services. However, there is lack of data to show increase use of private sectors. Available data mainly showed public/state clinics were mostly used. This is addressed in the review. 

5. The evolution of SRH in education curriculums and the ‘family life’ model drawing from the Demographic and Health Surveys ‘empowerment’ findings and teasing out women’s limited autonomy over decision making to name but a few.

However, if as it seems, this information wasn’t gleaned from the search results, then some more general references to these broader issues could be incorporated in the narrative. Page 5, paragraph 1, under “introduction”. Under “results – family planning service in the pacific”, page 25-27,These issues were considered as important and incorporated in the introduction and results section Issues were addressed in the manuscript 

6. It would be good if the references to the dates of the publications ‘published between 1994 and current’, ‘published between 1994 and 2019’ used in the paper, are more consistent and more precise in terms of ‘published between 1994 and < > 2019’. Page. 7, number 4. The publication dates for the review was changed to “1994 and 2019” The change was made from “current” to “2019” 

Reviewer 3 

1. It would be good to note that not all of the Pacific Island Countries are equally in their development status and a history of conflict in countries like PNG and Solomon Islands, which delay progress towards development. Page 4, paragraph 2, under “introduction”. This is noted as important information is included in the introduction of the review This is addressed in the revision

2. In the early part of the paper, it may be useful to utilise data on the gaps ie unmet need and contraceptive prevalence rates in the PICTs to illustrate the lack of uptake, and hence the need for this paper. This data is available through reports such as the Demographic Health Surveys and the World Contraceptive Use Reports by the UN Population Division Under “introduction”, page 5, paragraph 2, the data for “contraceptive prevalence rate” and “unmet need” for family planning, were included References to CPR and unmet need for family planning were inserted in the introduction

3. The effect of culture and religion in the Pacific Islands on issues of sexuality and reproduction, and women’s rights cannot be underestimated. This issue is critically reviewed to ensure a balanced view on culture and religion based on the data is analysed Addressed in the narrative 

4. In the introduction, the authors write that in response to China’s practices of coercive contraception and abortion, the Holy See issued a strong stance to these to these issues – (In response to the controversy, the Roman Catholic Church prohibited all artificial contraception and made a strong stand against abortion as a method of family planning [11]). This is inaccurate as: 1) The Holy See has had centuries-long tussle on the issue of abortion….

2) The Catholic Church’s stance against abortion is not only as a “method of family planning” but for even therapeutic reasons – rape, birth defects etc. The introduction is revised. The understanding of this issue is corrected in the narrative The issue is addressed and information updated

Kind regards,

Relmah Harrington (Corresponding Author)

---

## [Decision Letter · Decision Letter 1]

12 Jul 2021

Family planning in Pacific Island Countries and Territories (PICTs): A scoping review

PONE-D-20-25718R1

Dear Dr. Harrington,

We’re pleased to inform you that your manuscript has been judged scientifically suitable for publication and will be formally accepted for publication once it meets all outstanding technical requirements.

Kind regards,

José Antonio Ortega, Ph.D.

Academic Editor

PLOS ONE

Additional Editor Comments (optional):

Congratulations on a much improved manuscript!

While both reviewers indicate a minor revision, it is felt that the suggested very minor issues can be dealt with without the need for a subsequent round of revisions.

As reviewer 1 states, before publication please ensure to be systematic in the way contraceptive prevalence is reported for survey data in table 1, including always the rate for all women and for women in union (or married, depending on the survey definition). The rates provided now do not have a qualification and are not systematic making them useless.

See also a minor typo also pointed out by reviewer 1.

Reviewers' comments:

Reviewer's Responses to Questions

**Comments to the Author**

1. If the authors have adequately addressed your comments raised in a previous round of review and you feel that this manuscript is now acceptable for publication, you may indicate that here to bypass the “Comments to the Author” section, enter your conflict of interest statement in the “Confidential to Editor” section, and submit your "Accept" recommendation.

Reviewer #1: (No Response)

Reviewer #2: All comments have been addressed

2. Is the manuscript technically sound, and do the data support the conclusions?

Reviewer #1: Yes

Reviewer #2: Yes

3. Has the statistical analysis been performed appropriately and rigorously? 

Reviewer #1: N/A

Reviewer #2: N/A

4. Have the authors made all data underlying the findings in their manuscript fully available?

Reviewer #1: Yes

Reviewer #2: Yes

5. Is the manuscript presented in an intelligible fashion and written in standard English?

Reviewer #1: Yes

Reviewer #2: Yes

6. Review Comments to the Author

Reviewer #1: Result section- Page 25, sub-title “Family planning services in the Pacific”. Please correct “Thirteen papers [5, 44, 52, 56, 61, 64-67, 70, 74, 77] described government…”. There are only 12 references. This should be 13 references

Reviewer #2: Authors have made revisions to address reviewer comments given the research methods and criteria etc with the limitations of the method being very clear.

7. PLOS authors have the option to publish the peer review history of their article (what does this mean?). If published, this will include your full peer review and any attached files.

Reviewer #1: No

Reviewer #2: No

---

## [Editor Report · Acceptance letter]

26 Jul 2021

PONE-D-20-25718R1 

Family planning in Pacific Island Countries and Territories (PICTs): A scoping review 

Dear Dr. Harrington:

I'm pleased to inform you that your manuscript has been deemed suitable for publication in PLOS ONE. Congratulations! Your manuscript is now with our production department. 

Kind regards, 

on behalf of

Dr. José Antonio Ortega 

Academic Editor

PLOS ONE